# SafeMoE: Safe Fine-Tuning for MoE LLMs by Aligning Harmful Input Routing

**Jaehan Kim, Minkyoo Song, Seungwon Shin, Sooel Son**[*]
KAIST
{jaehan,minkyoo9,claude,sl.son}@kaist.ac.kr

## Abstract

Recent large language models (LLMs) have increasingly adopted the Mixture-of-Experts (MoE) architecture for efficiency. MoE-based LLMs heavily depend on a superficial safety mechanism in which harmful inputs are routed safety-critical experts. However, our analysis reveals that routing decisions for harmful inputs drift significantly after fine-tuning, exposing a critical vulnerability to harmful fine-tuning (HFT) attacks. Existing defenses, primarily designed for monolithic LLMs, are less effective for MoE LLMs as they fail to prevent drift in harmful input routing. To address this limitation, we propose SafeMoE, a safe fine-tuning method tailored to MoE LLMs. SafeMoE directly mitigates routing drift by penalizing the gap between the routing weights of a fine-tuned model and those of the initial safety-aligned model, thereby preserving the safety-aligned routing of harmful inputs to safety-critical experts. Experiments on open-source MoE LLMs ranging from 7B to 141B parameters demonstrate that SafeMoE effectively mitigates HFT attacks, reducing the harmfulness score of OLMoE from 62.0 to 5.0, for example, while maintaining task utility within 1% degradation and incurring only 2% overhead. It significantly outperforms state-of-the-art defense methods for safeguarding LLM fine-tuning and remains effective in recent large-scale MoE LLMs such as gpt-oss and Llama 4. Our implementation is available at https://github.com/jaehanwork/SafeMoE.

## 1 Introduction

Mixture-of-Experts (MoE) (Shazeer et al., 2017) is a sparse model architecture that improves efficiency by dynamically routing inputs to a subset of expert layers, which have gained adoption for large language models (LLMs). Recent MoE-based LLMs, including gpt-oss (OpenAI, 2025), Llama 4 (Meta AI, 2025a), Qwen3 MoE (Qwen Team, 2025), and DeepSeek-R1 (Guo et al., 2025), have achieved surpassing performance on a wide range of challenging tasks, outperforming their monolithic counterparts. However, recent studies (Lai et al., 2025; Fayyaz et al., 2025) show that the safety of MoE LLMs heavily relies on certain *safety-critical* experts and intentionally manipulating routing decisions to disable these experts leads to significant increases in harmfulness. This superficial safety mechanism leaves MoE LLMs particularly susceptible to harmful fine-tuning (HFT) attacks (Qi et al., 2024; Yang et al., 2024; Zhan et al., 2024; Huang et al., 2024a; Wallace et al., 2025). These attacks are designed to compromise the safety of a target LLM by injecting only a limited number of harmful samples into the training dataset, rendering a practical yet severe threat to commercial LLM providers given the growing prevalence of fine-tuning API services (OpenAI, 2024b; Google, 2025).

Our systematic analysis of MoE LLMs uncovers a novel architectural vulnerability in their routing mechanism, which determines which expert layers should be activated for processing inputs. We find that routing decisions for harmful inputs drift substantially from those of the initial safety-aligned model under both harmful and benign fine-tuning, a phenomenon we term *safety routing drift*. This drift impedes the activation of safety-critical experts and thereby undermines the model's safety.

---

[*]Corresponding author

Given the reliance of MoE LLM safety on routing aligned toward these safety-critical experts, preserving the initial routing decisions of the safety-aligned models is important to safeguarding MoE LLMs against HFT attacks. However, existing defenses (Huang et al., 2025b; Li et al., 2025a; Lu et al., 2025) are primarily designed for the monolithic transformer architecture and over-

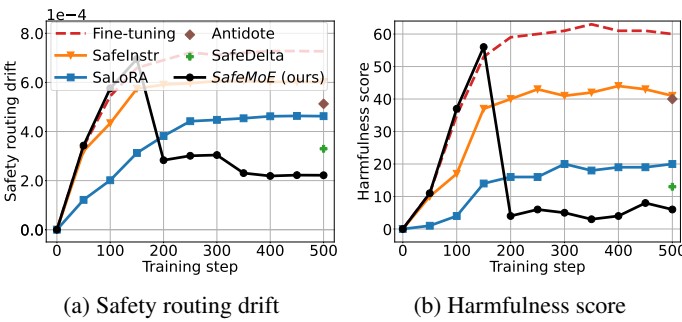

(a) Safety routing drift    (b) Harmfulness score

Figure 1: Effectiveness of defenses against HFT attacks.

look the superficial safety mechanisms of MoE LLMs, exhibiting limitations in mitigating fine-tuning risks. In support of this, we conduct a preliminary experiment on the training dynamics of existing defenses under HFT attacks, including fine-tuning-stage methods (Bianchi et al., 2024; Li et al., 2025a) and post-fine-tuning methods (Huang et al., 2025a; Lu et al., 2025), as shown in Figure 1. The safety routing drift is quantified as the deviation in routing weights on harmful instructions between the fine-tuned and safety-aligned models. In MoE LLMs, these state-of-the-art defenses become less effective in preventing the safety routing drift and reducing harmfulness of fine-tuned models.

To address this limitation, we propose SAFEMOE, the first safe fine-tuning method tailored to safeguard MoE-based LLMs, which directly addresses the vulnerability in their safety mechanisms. Specifically, we design a regularization technique that aligns the routing decisions of a fine-tuned MoE LLM with those of the initial safety-aligned model by minimizing the KL-divergence of their routing weights on harmful inputs during fine-tuning. This encourages the fine-tuned model to direct harmful inputs to safety-critical experts as in the safety-aligned model. Accordingly, the fine-tuned model withstands the effects of HFT attacks while attaining comparable task utility. To reduce the overhead of SAFEMOE, we adopt a greedy optimization strategy that alternates between the fine-tuning and regularization steps rather than optimizing both simultaneously.

We conduct extensive experiments across eight widely used MoE LLMs, ranging from 7B to 141B parameters. Our safety evaluation results demonstrate the surpassing effectiveness and robustness of SAFEMOE in safeguarding MoE LLMs against fine-tuning risks. For example, it effectively mitigates an HFT attack that raises the harmfulness score of OLMoE (7B) to 62.0, reducing it to 5.0 with only a 1% degradation in the task utility, outperforming state-of-the-art defenses significantly. This notable effectiveness remains consistent across diverse MoE LLMs, including larger and more advanced models such as gpt-oss and Llama 4, while maintaining their reasoning performance.

Based on an in-depth analysis of training dynamics, we confirm that our regularization technique is methodologically valid in preventing the safety routing drift and driving harmfulness reduction. We note that SAFEMOE is highly efficient, comparable to the baseline methods, with only approximately 2% training time overhead in both LoRA and full fine-tuning, which demonstrates its practicality to large-scale MoE LLMs. Through these findings, we highlight the importance of architecture-aware designs of safe fine-tuning methods for MoE LLMs.

Our contributions are summarized as follows:

- We identify a vulnerability in the safety mechanism of MoE LLMs, where the drift in routing decisions for harmful inputs during fine-tuning undermines their safety.
- We propose SAFEMOE, an effective and efficient safe fine-tuning method tailored to MoE LLMs that preserves the routing decisions of the initial safety-aligned model.
- Through experiments on open-source MoE LLMs, we show that this vulnerability is consistent across diverse models and that SAFEMOE offers robust mitigation against HFT attacks.

## 2  PRELIMINARIES

**Mixture-of-experts (MoE).** Figure 2 shows the standard MoE architecture for LLMs (Shazeer et al., 2017; Lepikhin et al., 2021; Du et al., 2022; Komatsuzaki et al., 2023; Fedus et al., 2022).

The MoE layer output is formalized as:

$$h_{\mathrm{MoE}} = \sum_{i \in N} \mathrm{TopK}(\sigma(\boldsymbol{r}(x)))\mathrm{FFN}_i(x), \qquad (1)$$

where $N$ is the number of experts, $\sigma$ is Softmax, $\boldsymbol{r}(h) \in \mathbb{R}^N$ is routing weights on an input token $x$, and $\mathrm{FFN}_i$ is the $i$-th expert layer. By leveraging this conditional computation approach, the MoE architecture provides substantial efficiency gains in both training and inference, which has driven its adoption in recent LLMs. MoE LLMs typically adhere to the transformer architecture, but the feed-

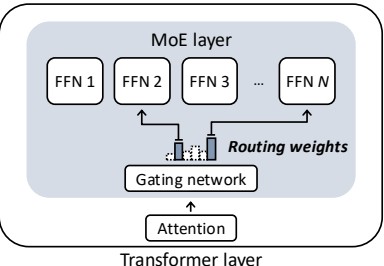

Figure 2: MoE LLM architecture.

forward network (FFN) in each transformer layer is replaced by an MoE layer. The MoE layer consists of multiple independent FFNs, referred to as *experts*, along with a gating network. For each input token, the gating network dynamically assigns a *routing weight* to every expert in the MoE layer based on the token's hidden state from the self-attention layer. The top-$k$ experts are then selected for forward computation, and their outputs are combined according to the assigned weights.

**Superficial safety mechanism in MoE.** Previous studies have examined safety vulnerabilities in LLMs by analyzing specific model parameters and forwarding paths (Wei et al., 2024; Lee et al., 2024; Peng et al., 2024; Tamirisa et al., 2025). However, the safety implications of the fundamentally different mechanisms introduced by the MoE architecture remain insufficiently understood. Routing decisions in MoE LLMs play a crucial role in composing the outputs and have a direct impact on their safety. Recent studies (Lai et al., 2025; Fayyaz et al., 2025) show that harmful instructions to safety-aligned MoE LLMs, such as *"How to make a bomb?"*, consistently trigger specific experts, referred to *safety-critical experts*. Moreover, masking these experts causes significant degradation in safety even when the model parameters remain unchanged. This indicates that the safety of MoE LLMs heavily depends on routing decisions that determine the activation of safety-critical experts.

**Harmful fine-tuning (HFT) attacks.** Safety alignment has become an essential step in ensuring that LLM outputs are harmless and aligned with human values (Ouyang et al., 2022; Rafailov et al., 2023). However, researchers show that this alignment can be undone through user fine-tuning (Qi et al., 2024; Yang et al., 2024; Zhan et al., 2024). Specifically, they find that HFT attacks that inject a small portion of harmful samples into the training dataset, or even benign fine-tuning alone, can severely impair LLM safety. Considering the increasing availability of fine-tuning API services (OpenAI, 2024b; Google, 2025), HFT attacks have become a practical threat to LLM providers.

To mitigate this threat, several methods have been proposed to enhance the safety of LLM fine-tuning—for example, augmenting datasets with safe samples (Bianchi et al., 2024; Zong et al., 2024), constraining training to prevent harmful-direction drift (Huang et al., 2024b; Wu et al., 2025), and pruning harmful parameters from the fine-tuned models (Huang et al., 2025a; Lu et al., 2025).

However, existing defenses focus solely on monolithic LLMs and overlook the distinct architecture of MoE LLMs based on dynamic input routing and their superficial safety mechanisms. To the best of our knowledge, no prior work has investigated safe fine-tuning strategies tailored to MoE LLMs.

## 3 SAFETY VULNERABILITY IN MoE LLMs

Given that the safety of MoE LLMs depends on safety-critical experts (Lai et al., 2025; Fayyaz et al., 2025), we posit that *safety degradation in fine-tuned MoE LLMs arises from substantial deviations in routing decisions for harmful instructions compared to those in the initial safety-aligned models.*

To verify this, we first define *safety routing drift* for harmful inputs. This metric quantifies a difference between the routing weights of a safety-aligned MoE LLM and its fine-tuned counterpart:

**Definition 3.1** (Safety routing drift). *Let $\boldsymbol{w}_{align}$ be a safety-aligned MoE LLM. Given a fine-tuned model $\boldsymbol{w}_{ft}$, the safety routing drift for a harmful instruction input $x$ is defined as:*

$$d(\boldsymbol{w}_{ft}, x) = D_{KL}\left(\sigma(\boldsymbol{r}(x|\boldsymbol{w}_{align})) \,\big|\big|\, \sigma(\boldsymbol{r}(x|\boldsymbol{w}_{ft}))\right), \qquad (2)$$

*where $D_{KL}(P||Q)$ denotes KL divergence between a reference distribution $P$ and an approximating probability distribution $Q$, $\sigma$ is Softmax, and $\boldsymbol{r}(\cdot|\boldsymbol{w}) \in \mathbb{R}^N$ denotes the routing weight vector over $N$ experts of model $\boldsymbol{w}$ for an input.*

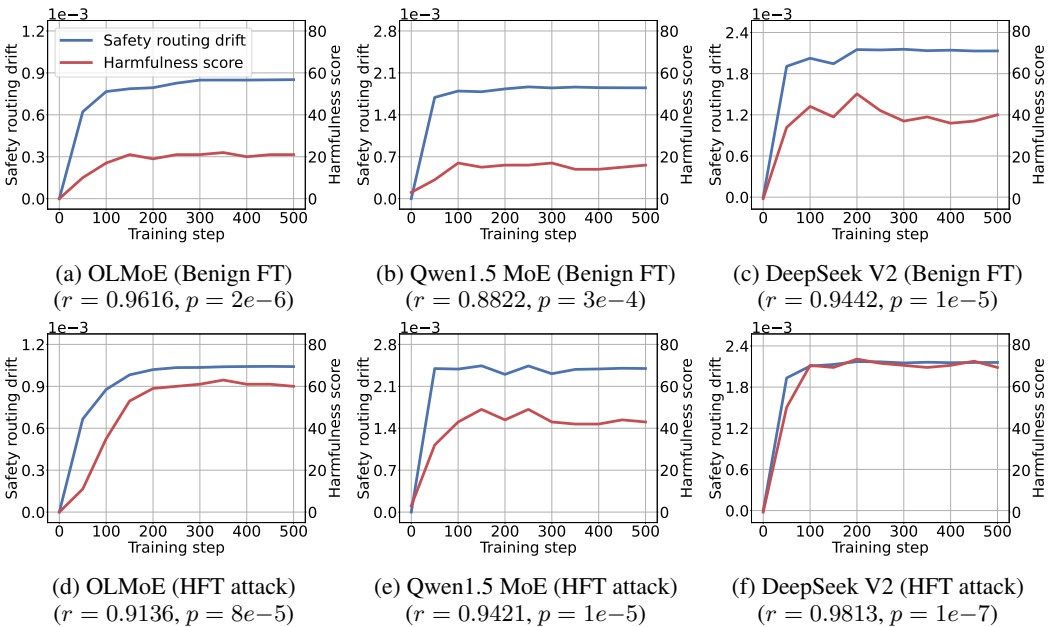

(a) OLMoE (Benign FT)
$(r = 0.9616, p = 2e{-}6)$

(b) Qwen1.5 MoE (Benign FT)
$(r = 0.8822, p = 3e{-}4)$

(c) DeepSeek V2 (Benign FT)
$(r = 0.9442, p = 1e{-}5)$

(d) OLMoE (HFT attack)
$(r = 0.9136, p = 8e{-}5)$

(e) Qwen1.5 MoE (HFT attack)
$(r = 0.9421, p = 1e{-}5)$

(f) DeepSeek V2 (HFT attack)
$(r = 0.9813, p = 1e{-}7)$

Figure 3: Safety routing drift and harmfulness of MoE LLMs over training steps. Results of t-tests for Pearson correlation coefficients are reported ($r$: correlation coefficient, $p$: $p$-value).

**Safety routing drift during fine-tuning.** We then analyze the training dynamics of three MoE LLMs, including OLMoE (Muennighoff et al., 2025), Qwen1.5 MoE (Qwen, 2024), and DeepSeek V2 (Liu et al., 2024), and measure the correlation between the safety routing drift and harmfulness of their fine-tuned models. Specifically, we consider two scenarios: i) benign fine-tuning on 5.5k samples from the Alpaca dataset (Taori et al., 2023), and ii) HFT on a combined dataset of 5k task-specific samples from SAMSum (Gliwa et al., 2019) and 500 harmful samples from BeaverTails (Ji et al., 2023). For each fine-tuned LLM, we compute safety routing drift on the last tokens of harmful instructions from JailbreakBench (Chao et al., 2024). Harmfulness scores are computed as the proportion of *unsafe* responses, evaluated using Llama-Guard-4-12B (Meta AI, 2025b).

Figure 3 presents the results of our analysis. In both fine-tuning scenarios, we observe significant safety routing drift from the initial safety-aligned model, with the drift metrics increasing as training progresses. Notably, routing decisions for harmful instructions drift even under benign fine-tuning, causing nontrivial increases in harmfulness. This demonstrates that the superficial safety mechanism of MoE LLMs is highly fragile and easily disrupted by fine-tuning. Moreover, the magnitude of safety routing drift is strongly correlated with harmfulness scores across the models, with high statistical significance ($p \ll 0.05$). Harmful fine-tuning (HFT) attacks even further amplify this drift, producing substantially larger increases in harmfulness than benign fine-tuning.

**Safety recovery through initial routing override.** To design an effective defense, we evaluate fine-tuned MoE LLMs by overriding their routing decisions, specifically by restoring the routing weights used for harmful inputs to those of the initial safety-aligned model during inference. Figure 4 presents the safety evaluation results for the OLMoE model fine-tuned on the SAMSum dataset. While fine-tuned models exhibit substantial harmfulness, overriding the initial routing decisions significantly reduces harmful outputs. This observation further highlights the critical role of the routing mechanism in MoE safety and demonstrates that preserving safety-aligned routing behaviors is a promising strategy for mitigating safety risks incurred during fine-tuning.

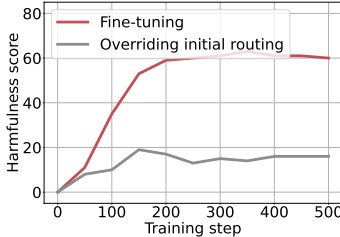

Figure 4: Harmfulness when overriding the safety-aligned routing decisions (OLMoE).

**Motivation for an MoE-specific defense.** Our analysis shows that the safety routing drift is highly correlated with the safety degradation in fine-tuned MoE LLMs. This suggests the need for an

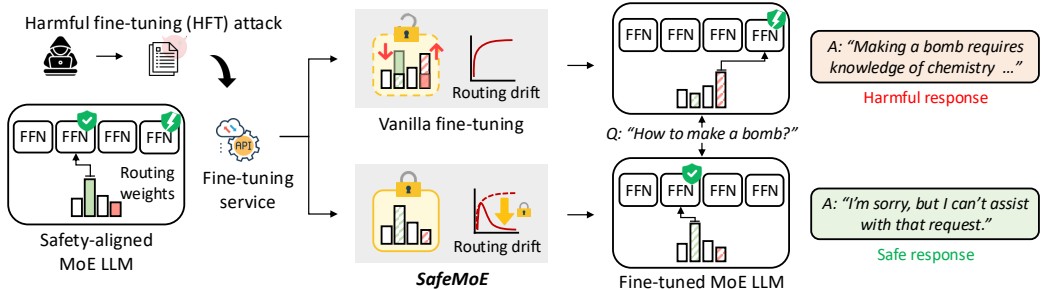

Figure 5: Overview of SAFEMOE. It mitigates the safety routing drift by directly constraining this drift during fine-tuning, thereby effectively safeguarding MoE LLMs against HFT attacks.

approach that preserves the initial routing decisions of the safety-aligned models on harmful inputs to mitigate fine-tuning risks in MoE LLMs. However, prior defenses, designed for monolithic LLMs, are limited in preventing the safety routing drift and reducing the harmfulness of fine-tuned MoE LLMs, as shown in our preliminary experiments (see Figure 1). These findings motivate a new defense that reflects the unique architecture of MoE LLMs and their safety mechanism.

## 4 SAFE MOE FINE-TUNING METHOD

**Safety routing drift regularization.** We propose a novel fine-tuning approach designed to improve the safety of fine-tuned MoE LLMs, as illustrated in Figure 5. Specifically, during fine-tuning, we propose leveraging a regularization objective that constrains the safety routing drift (Definition 3.1). This objective penalizes deviations in the routing weight distributions of an MoE LLM $\boldsymbol{w}$ under fine-tuning from those of the initial safety-aligned model $\boldsymbol{w}_{align}$ on harmful instructions:

$$\mathcal{L}_{reg}(\boldsymbol{w}) = \mathbb{E}_{x \in \mathcal{D}_h} \mathbb{E}_{l \in L} D_{\mathrm{KL}} \left( \sigma(\boldsymbol{r}^{(l)}(x|\boldsymbol{w}_{align})/\tau) \,\big|\big|\, \sigma(\boldsymbol{r}^{(l)}(x|\boldsymbol{w})/\tau) \right), \quad (3)$$

where $\mathcal{D}_h$ is the harmful instruction dataset, and $L$ is the set of transformer layers. We apply regularization to the routing weights assigned to the last token of each harmful instruction. The temperature $\tau$ controls the strength of regularization. A smaller $\tau$ (e.g., $< 1.0$) further enhances safety by sharpening the routing weight distribution, which enables the regularization to focus more on top-ranked safety-critical experts and tightly constrains the routing weights of those experts.

**Bi-level greedy optimization**. We integrate the regularization into supervised fine-tuning on a task-specific dataset $\mathcal{D}_{ft}$ to solve:

$$\arg\min_{\boldsymbol{w}} \ \mathcal{L}_{\mathrm{sft}}(\boldsymbol{w}) + \mathcal{L}_{\mathrm{reg}}(\boldsymbol{w}). \quad (4)$$

Simultaneously optimizing both losses at every training step, however, incurs substantial computational overhead. To address this, we reframe the joint optimization into a bi-level greedy approach that alternates between the supervised fine-tuning and regularization steps.

The greedy optimization process is described in Algorithm 1. We first precompute the routing weights of the initial safety-aligned MoE LLM over all harmful instructions in $\mathcal{D}_h$ to avoid redundant forward passes in the regularization steps (line 2). During training with the fine-tuning loss $\mathcal{L}_{sft}$, we insert safety-preserving steps using the regularization loss $\mathcal{L}_{reg}$ every $T_{reg}$ steps (line 6).

## 5 EXPERIMENTS

### 5.1 SETUP

**MoE LLMs.** We conduct experiments on eight widely adopted MoE LLMs: OLMoE-1B-7B-0125-Instruct (Muennighoff et al., 2025), Qwen1.5-MoE-A2.7B-Chat (Qwen, 2024), and DeepSeek-V2-Lite-Chat (Liu et al., 2024), gpt-oss-20b (OpenAI, 2025), Qwen3-30B-A3B (Qwen Team, 2025), Phi-3.5-MoE-Instruct (Microsoft, 2024), and two models with on-the-fly 4-bit quantization, Llama-4-Scout-17B-16E-Instruct (Meta AI, 2025a) and Mixtral-8x22B-Instruct-v0.1 (Mistral AI, 2025). All models used are safety-aligned versions. We describe model details in the Appendix A.2.

Table 1: Safety evaluation of MoE LLMs. We report fine-tuning accuracy (FA↑) and harmfulness score (HS↓) for the SAMSum and SQL tasks. The number of parameters is denoted as (active/total).

| Method | OLMoE (1.3B/6.9B) | | | | Qwen1.5 MoE (2.7B/14.3B) | | | | DeepSeek V2 (2.4B/15.7B) | | | |
|---|---|---|---|---|---|---|---|---|---|---|---|---|
| | SAMSum | | SQL | | SAMSum | | SQL | | SAMSum | | SQL | |
| | FA | HS | FA | HS | FA | HS | FA | HS | FA | HS | FA | HS |
| Aligned | 31.8 | 0 | 43.0 | 0 | 36.6 | 3.0 | 38.4 | 3.0 | 34.0 | 0 | 29.6 | 0 |
| Fine-tuning | 49.3 | 62.0 | 58.5 | 64.0 | 50.4 | 49.0 | 70.2 | 37.0 | 52.0 | 70.0 | 70.1 | 72.0 |
| SafeInstr | **49.5** | 46.0 | 58.9 | 41.0 | 50.6 | 11.0 | 69.3 | 8.0 | **52.1** | 25.0 | **70.2** | 21.0 |
| Lisa | 48.4 | 21.0 | 57.2 | 40.0 | 50.1 | 10.0 | 68.5 | 13.0 | 50.7 | 24.0 | 68.9 | 22.0 |
| SaLoRA | 48.9 | 24.0 | 54.5 | 40.0 | 48.9 | 28.0 | 54.9 | 33.0 | 50.1 | 66.0 | 62.0 | 74.0 |
| Antidote | 48.7 | 40.0 | 57.5 | 44.0 | 49.3 | 18.0 | 68.6 | 29.0 | 50.7 | 70.0 | 65.3 | 62.0 |
| SafeDelta | 48.6 | 13.0 | 57.4 | 33.0 | 50.2 | 22.0 | 69.2 | 30.0 | 51.0 | 47.0 | 69.0 | 72.0 |
| SAFEMOE | 48.9 | **5.0** | **59.0** | 17.0 | 50.6 | **0** | **69.5** | **1.0** | 51.0 | **1.0** | 69.1 | **4.0** |

**Fine-tuning datasets.** We consider two tasks: SAMSum (Gliwa et al., 2019) for dialogue summarization and SQL (b-mc2, 2023) for SQL query generation, both widely adopted for implementing HFT attack scenarios (Yang et al., 2025a; Wang et al., 2024). For effective attacks, we conduct supervised fine-tuning with 5k task samples combined with 500 harmful samples from BeaverTails (Ji et al., 2023). Fine-tuning is performed with LoRA (Hu et al., 2022), as detailed in Appendix A.2.

**Metrics.** We reports two metrics: *Fine-tuning accuracy (FA)* and *Harmfulness score (HS)*. FA measures task utility—Rouge-1 score for SAMSum and exact match accuracy for SQL. To assess reasoning performance, we measure accuracy on 570 samples (10 from each categories) from MMLU-Redux-2.0 (Gema et al., 2025). HS refers to the proportion of responses to JailbreakBench (Chao et al., 2024) instructions that are classified as *unsafe* by Llama-Guard-4-12B (Meta AI, 2025b).

**Baselines.** We evaluate five state-of-the-art defenses against HFT attacks. Among fine-tuning-stage methods, SafeInstr (Bianchi et al., 2024) augments fine-tuning datasets with additional safe samples, Lisa (Huang et al., 2024b) introduces a proximal term to prevent excessive drift in model parameters during fine-tuning, and SaLoRA (Li et al., 2025a) initializes LoRA layers with weights optimized on safe samples. We also consider two post-fine-tuning weight modification approaches. Antidote (Huang et al., 2025a) prunes harmful parameters by analyzing their contribution to harmful instructions. SafeDelta (Lu et al., 2025) selects delta parameters that maximize utility while minimizing safety degradation.

**SAFEMOE implementation.** We use 100 harmful instruction samples from SafeInstr (Bianchi et al., 2024) as the dataset $\mathcal{D}_h$. We select $\tau$ as the smallest value in the range 0.1-1.3 that allows a 1% degradation in fine-tuning accuracy. By default, $T_{\text{reg}}$ is set to the number of steps per epoch.

## 5.2 SAFETY EVALUATION

Table 1 reports the defense effectiveness of SAFEMOE on three widely used MoE LLMs. The initial safety-aligned models prior to fine-tuning (first row) are highly safe, exhibiting nearly zero harmfulness scores. Vanilla fine-tuning substantially undermines safety while improving task performance (second row). SAFEMOE attains the lowest harmfulness score while incurring only minimal loss in fine-tuning accuracy. For example, it reduces the harmfulness score of DeepSeek V2 fine-tuned on the SQL task from 72.0 to 4.0. SAFEMOE's superior effectiveness is attributed to its MoE-specific design, which explicitly encourages the routing of harmful instructions to safety-critical experts.

In contrast, baseline methods fail to effectively mitigate safety degradation. SafeInstr, which adopts an architecture-agnostic strategy based on safety data augmentation, achieves moderate defense performance but still leaves substantial harmfulness. Lisa also falls short in defending MoE LLMs against HFT attacks, highlighting the effectiveness of explicitly constraining drift in routing weights rather than in model parameters. SaLoRA and post-fine-tuning methods (Antidote and SafeDelta) assume all parameters are always activated, as in monolithic LLMs. In MoE LLMs, however, the dynamically changing active parameters hinder full optimization of these methods. Furthermore,

Table 2: Extended safety evaluation on five large-scale MoE LLMs. We report reasoning performance on MMLU-Redux-2.0 (MMLU↑) and harmfulness score (HS↓).

| Method | gpt-oss (3.6B/20.9B) | | Qwen3 MoE (3.3B/30.5B) | | Phi 3.5 MoE (6.6B/41.9B) | | Llama 4 (17B/109B) | | Mixtral (39B/141B) | |
|---|---|---|---|---|---|---|---|---|---|---|
| | MMLU | HS | MMLU | HS | MMLU | HS | MMLU | HS | MMLU | HS |
| Aligned | 85.4 | 2.0 | 89.6 | 1.0 | 83.3 | 2.0 | 90.4 | 7.0 | 78.9 | 7.0 |
| Fine-tuning | 77.5 | 84.0 | 89.1 | 67.0 | 80.7 | 83.0 | 89.5 | 79.0 | 66.5 | 78.0 |
| SAFEMOE | 79.6 | 7.0 | 88.8 | 4.0 | 81.4 | 2.0 | 89.8 | 3.0 | 78.4 | 8.0 |

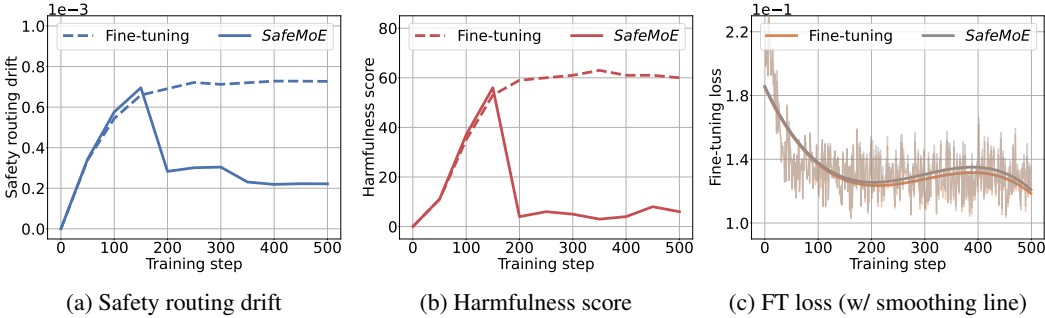

(a) Safety routing drift    (b) Harmfulness score    (c) FT loss (w/ smoothing line)

Figure 6: Training dynamics of vanilla fine-tuning vs. SAFEMOE (OLMoE on SAMSum).

they cause significant degradation in fine-tuning accuracy with only marginal harmfulness reduction, even though we extensively tune their hyperparameters (see Appendix A.3).

These results highlight the importance of architecture-aware defenses that directly address vulnerabilities in the safety mechanism. In Qwen1.5 MoE, we observe slightly improved safety compared to the safety-aligned models. This effect appears to result from the generalization ability of SAFEMOE, which promotes stronger activation of top-ranked safety-critical experts (see Appendix A.5).

**Results on larger MoE LLMs.** We further evaluate SAFEMOE on recent larger MoE LLMs under a strong HFT attack scenario using 500 purely harmful samples (Bianchi et al., 2024; Lu et al., 2025; Hsu et al., 2024), as shown in Table 2. These models employ diverse MoE configurations (e.g, activating 1 expert among 16 in Llama 4 and activating 8 among 128 in Qwen3 MoE) and distinct reasoning strategies (e.g., multi-level reasoning in gpt-oss and the thinking mode in Qwen3 MoE). Despite their differences, SAFEMOE generally achieves strong defense performance while effectively preserving the models' reasoning capability. For gpt-oss, Phi 3.5 MoE, and Mixtral, SAFEMOE even alleviates the degradation of reasoning performance observed in vanilla fine-tuning by preventing excessive overfitting to the HFT attack data. Overall, these results demonstrate the practicality of SAFEMOE for real-world fine-tuning services with high-capable MoE LLMs.

## 5.3    TRAINING DYNAMICS

To demonstrate that the design of SAFEMOE is valid in practice, we analyze the variation of the safety routing drift, harmfulness score, and fine-tuning loss throughout the fine-tuning steps. Figure 6 presents the results for OLMoE fine-tuned on the SAMSum task.

As shown in Figure 6a and 6b, SAFEMOE moderately reduces the safety routing drift at the early stage and effectively mitigates it after the first regularization period (after 150 steps). The harmfulness score decreases with drift mitigation, achieving a significant reduction compared to vanilla fine-tuning at subsequent checkpoints. Our experiment has shown that existing defenses have limited effectiveness in reducing the safety routing drift and harmfulness (see Figure 11). SAFEMOE methodologically addresses this limitation and provides an effective defense for MoE LLMs.

In Figure 6c, we also present the fine-tuning loss to further evaluate its impact on the fine-tuning task. The loss converges stably, closely matching the trajectory of vanilla fine-tuning with only

Table 3: Execution overheads (OLMoE).

| Method | Extra time | |
|---|---|---|
| | Seconds | Percentage |
| SafeInstr | 15.98 | 1.98% |
| SaLoRA | 747.56 | 92.41% |
| Antidote | 5.67 | 0.70% |
| SafeDelta | 52.18 | 6.45% |
| SAFEMOE | 17.26 | 2.13% |

Table 4: Ablation study on restricting gradients from the expert layers. We report fine-tuning accuracy (FA↑) and harmfulness score (HS↓).

| Method | OLMoE on SAMSum | |
|---|---|---|
| | FA | HS |
| Fine-tuning | 49.3 | 62.0 |
| SAFEMOE | 48.9 | 5.0 |
| SAFEMOE (ablation) | 48.2 | 18.0 |

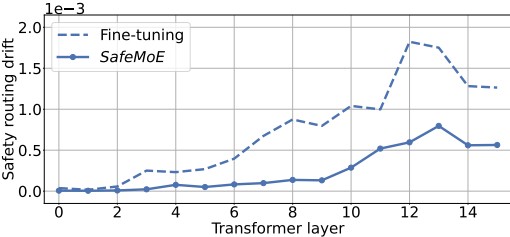

Figure 7: Safety routing drift across transformer layers (OLMoE on SAMSum).

Figure 8: Effectiveness of SAFEMOE applied on specific layers (OLMoE on SAMSum).

negligible differences. This underscores the compatibility of SAFEMOE in standard fine-tuning, enabling safe and practical fine-tuning services.

## 5.4 EFFICIENCY ANALYSIS

To assess the efficiency of SAFEMOE, we analyze the total execution overhead of SAFEMOE during fine-tuning and post-fine-tuning. Table 3 summarizes results for OLMoE fine-tuned on the SAMSum task under the environment described in Appendix A.2. For reference, vanilla fine-tuning takes 808.98 seconds on average over ten trials. SaLoRA incurs substantial overhead in computing the optimal LoRA initialization, increasing the execution time to nearly twice that of vanilla fine-tuning. The other approaches are generally more efficient but fail to effectively mitigate HFT attacks on MoE LLMs. In contrast, SAFEMOE attains strong safety with only a 2.13% increase in its training time, highlighting its practicality in safeguarding large-scale models.

## 5.5 ABLATION STUDY

**Isolation of SAFEMOE's influence on routing.** In the MoE architecture, the embedding vector transferred from the attention layer feeds into both the gating network and the expert layers. To isolate the effect of SAFEMOE on the gating network, we explore an ablation in which gradients from the expert layers are blocked when updating the embedding vector. As shown in Table 4, this ablation setting still achieves strong defense performance, validating our approach of directly preserving routing weights for harmful inputs.

**Layer-selective application of SAFEMOE.** We analyze the extent of safety routing drift across transformer layers in OLMoE fine-tuned on the SAMSum task, as shown in Figure 7. The results reveal that routing drift is not uniform across layers and the upper layers exhibit substantially larger drift. This appears to result from harmful features typically being distinguished after the middle layers of LLMs (Li et al., 2025b).

Motivated by this observation, we investigate more efficient variants of SAFEMOE that apply the safety routing drift regularization selectively rather than across all layers (Figure 8). The safety evaluation results show that targeting upper layers (e.g., 12-15 layers) provides much stronger mitigation of HFT attacks than targeting lower layers. Moreover, applying SAFEMOE only to 8-15 layers (the upper half) achieves a level of safety comparable to full-layer regularization.

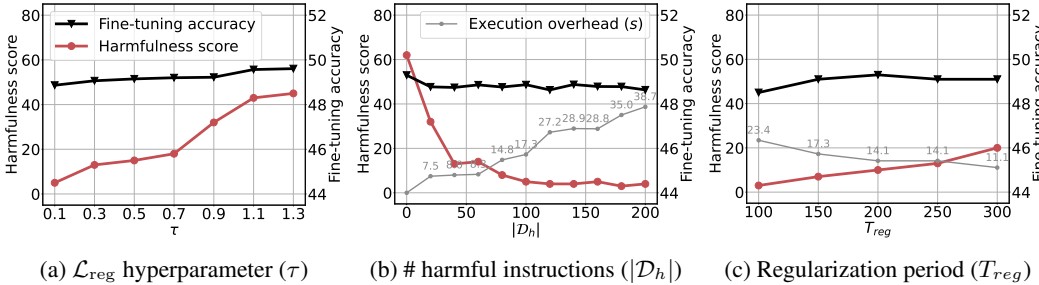

(a) $\mathcal{L}_{\text{reg}}$ hyperparameter ($\tau$)     (b) # harmful instructions ($|\mathcal{D}_h|$)     (c) Regularization period ($T_{reg}$)

Figure 9: Sensitivity to regularization strength (OLMoE on SAMSum).

## 5.6 SENSITIVITY ANALYSIS

**Regularization hyperparameter.** The hyperparameter $\tau$ in Equation 3 controls the strength of the safety routing drift regularization. Figure 9a shows results for OLMoE fine-tuned with the SAMSum task. Smaller values of $\tau$ focus the regularization on top-ranked safety-critical experts, achieving substantial harmfulness reduction. Across $\tau$ values, fine-tuning accuracy remains nearly unchanged, demonstrating that SAFEMOE can robustly improve safety without sacrificing task utility.

**Number of harmful instructions.** We vary the number of harmful instructions $|\mathcal{D}_h|$ used for the regularization (see Figure 9b). More harmful instructions strengthen the safety effect with an approximately linear overhead increase, suggesting a simple yet effective way to enhance our approach. Our default choice ($|\mathcal{D}_h| = 100$) provides a practical balance between safety and efficiency.

**Regularization period.** We examine the influence of the regularization frequency by varying the regularization period $T_{reg}$, as shown in Figure 9c. A smaller $T_{reg}$ triggers more frequent drift regularization steps, resulting in stronger defense performance but incurring additional execution-time overhead. Despite this trade-off between efficiency and safety, we emphasize that the overhead remains reasonable and acceptable, when compared to the baseline methods discussed in Section 5.4.

**Harmful sample ratio in fine-tuning data.** We explore different strengths of HFT attacks by adjusting the ratio of harmful samples in the fine-tuning dataset $\mathcal{D}_{ft}$ (see Figure 10). Vanilla fine-tuning exhibits drastically increasing harmfulness scores as the harmful ratio rises. In contrast, SAFEMOE suppresses harmfulness escalation, demonstrating robustness even against stronger attacks.

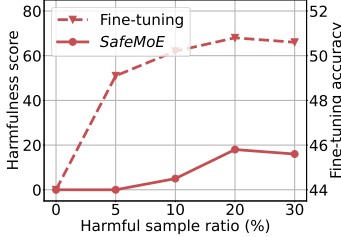

Figure 10: Sensitivity to the harmful sample ratio in $\mathcal{D}_{ft}$.

## 5.7 EFFECTIVENESS UNDER FULL FINE-TUNING

We extend our evaluation to a full-parameter fine-tuning scenario to demonstrate the generality of SAFEMOE. Table 5 shows the results for OLMoE fine-tuned on the SAMSum task. SAFEMOE achieves substantial harmfulness reduction while preserving fine-tuning accuracy in this setting as well. Although all expert layers, including safety-critical ones, are exposed to training during HFT attacks, SAFEMOE remains robustly effective by solely preventing the safety routing drift. This further highlights the central role of routing in the safety of MoE LLMs and confirms the validity of our approach. Moreover, it incurs only a 2.30% increase in execution time (four GPUs), comparable to that observed in the LoRA-based setting (2.13% with one GPU). These results show that SAFEMOE is reliably adaptable to both parameter-efficient and full fine-tuning approaches.

## 5.8 EVALUATION ON ADDITIONAL HARMFULNESS BENCHMARK

To further validate our findings, we evaluate harmfulness of fine-tuned MoE LLMs on HEx-PHI (Qi et al., 2024), a widely used harmful instruction benchmark. Table 6 reports results for the SAMSum fine-tuning scenario under our main settings. The results remain consistent, showing that SAFEMOE significantly outperforms all baselines, while SafeInstr achieves moderate harmfulness reduction

Table 5: Full fine-tuning results of fine-tuning accuracy (FA↑), harmfulness score (HS↓), and training time.

| Method | OLMoE on SAMSum | | |
| --- | --- | --- | --- |
| | FA | HS | Time ($s$) |
| Aligned | 31.8 | 0 | - |
| Fine-tuning | 50.4 | 58.0 | 7,023.79 |
| SAFEMOE | 51.0 | 2.0 | 7,189.01 (+2.30%) |

Table 6: Harmfulness score (HS↓) on HEx-PHI.

| Method | OLMoE | Qwen1.5 MoE | DeepSeek V2 |
| --- | --- | --- | --- |
| Aligned | 0.3 | 8.7 | 5.7 |
| Fine-tuning | 79.7 | 33.0 | 83.0 |
| SafeInstr | 52.0 | 13.3 | 31.0 |
| SaLoRA | 39.3 | 34.7 | 64.7 |
| Antidote | 53.3 | 24.7 | 68.0 |
| SafeDelta | 21.7 | 28.7 | 41.0 |
| SAFEMOE | 6.3 | 10.0 | 3.3 |

with strong performance for Qwen1.5 MoE. These findings confirm the robust defense effectiveness of SAFEMOE against diverse harmful querying scenarios.

## 6 RELATED WORK

We categorize recent defenses against HFT attacks into three groups based on their application stage.

**Alignment stage.** These methods aim to enhance the robustness of the model against subsequent HFT attacks. Vaccine (Huang et al., 2024c) and its memory-efficient variant (Liu et al., 2025) trains models to resist perturbations that maximize alignment loss. RepNoise (Rosati et al., 2024) and Booster (Huang et al., 2025b) proactively remove harmful information by optimizing perturbations in model representations or weights. VAA (Liang et al., 2025) introduces a vulnerability-aware alignment method that balances training across vulnerable and invulnerable subsets of alignment data. However, such alignment-stage methods require carefully tuned hyperparameter for each downstream task (Huang et al., 2025b), which limits their practicality in fine-tuning services that must handle many unknown tasks.

**Fine-tuning stage.** A second line of work directly addresses safety degradation during fine-tuning. SafeInstr (Bianchi et al., 2024) augments supervised fine-tuning datasets with safe samples. Lisa (Huang et al., 2024b) introduces a proximal optimization method to mitigate convergence instability when jointly training alignment and task-specific data. AsFT (Yang et al., 2025b) adopts a regularization term to suppress updates in harmful directions. SAFT (Choi et al., 2024) and SEAL (Shen et al., 2025) identify and filter harmful samples from fine-tuning data by scoring their safety impact. SaLoRA (Li et al., 2025a) proposes a safety-aware initialization of LoRA layers, designed based on an analysis of changes in safety-related features observed during fine-tuning.

**Post-fine-tuning stage.** Training-free remedies have also been proposed to restore safety after harmful fine-tuning. RESTA (Bhardwaj et al., 2024) extracts a safety vector from an aligned model and reintroduces it into the fine-tuned model via arithmetic addition of weights. SafeLoRA (Hsu et al., 2024) selectively projects LoRA weights into a safety-aligned subspace. Antidote (Huang et al., 2025a) removes harmful parameters identified through their importance to alignment data, thereby improving robustness under varying fine-tuning hyperparameters. SafeDelta (Lu et al., 2025) refines fine-tuned delta parameters to balance task utility with reduced safety degradation.

## 7 CONCLUSION

This work introduces SAFEMOE, the first safe fine-tuning method tailored to MoE-based LLMs. Our systematic analysis uncovers a vulnerability inherent in their safety mechanisms; routing decisions for harmful inputs drift significantly from those of safety-aligned models under both harmful and benign fine-tuning. To address this, we propose a routing drift regularization method with an efficient optimization algorithm that integrates seamlessly into standard MoE LLM fine-tuning pipelines. Extensive evaluations across diverse MoE LLMs show that SAFEMOE achieves significant reductions in harmfulness with minimal overhead. These results establish SAFEMOE as an effective and practical defense for fine-tuning services against HFT attacks, underscoring the need to address architectural weakness when safeguarding MoE LLMs from fine-tuning risks.

ETHICS STATEMENT

This work presents a safe fine-tuning method designed to mitigate potential safety threats associated with LLM fine-tuning. In our experiments, we evaluate the extent of harmful behaviors exhibited by open-source LLMs under harmful fine-tuning (HFT) attacks. HFT attacks are a well-documented concern (Qi et al., 2024; Yang et al., 2024; Zhan et al., 2024; Bianchi et al., 2024; Zong et al., 2024; Huang et al., 2024b; Li et al., 2025a), and our experiments are conducted entirely using publicly available alignment datasets and benchmarks related to safety and HFT attacks. Accordingly, we believe our work does not introduce any additional harm. Furthermore, we ensure that neither the harmful LLMs nor their generated harmful responses are shared, strictly adhering to the ICLR Code of Ethics (ICLR, 2025) throughout the work.

REPRODUCIBILITY STATEMENT

We release the implementation of SAFEMOE to facilitate reproduction our method and evaluation results. The source code is available at `https://github.com/jaehanwork/SafeMoE`

THE USE OF LARGE LANGUAGE MODELS

The authors used ChatGPT (OpenAI, 2025) for grammatical refinements of the manuscript. These modifications have been manually reviewed and finalized by the authors.

ACKNOWLEDGMENT

This work was partly supported by the Institute of Information & communications Technology Planning & Evaluation (IITP) grant funded by the Korea government (MSIT) (No.RS-2020-II200153, Penetration Security Testing of ML Model Vulnerabilities and Defense, 40%, No.RS-2024-00337703, Development of Satellite Security Vulnerability Detection Techniques Using AI and Specification-Based Automation Tools, 30%) and the InnoCORE program of the Ministry of Science and ICT (N10250156, 30%).

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

# A APPENDIX

## A.1 BI-LEVEL GREEDY OPTIMIZATION OF SAFEMOE

---
**Algorithm 1** Greedy optimization of safety routing drift regularization

---
**Input:** Safety-aligned MoE LLM $\boldsymbol{w}_{align}$; Fine-tuning datasets $\mathcal{D}_{ft}$; Harmful instruction dataset $\mathcal{D}_h$; Total training steps $T$; Regularization period $T_{reg}$; Optimizer $\text{Adam}(\eta, \beta_1, \beta_2, \epsilon)$
**Output:** The fine-tuned MoE LLM
 1: Initialize model weights $\boldsymbol{w}_0 \leftarrow \boldsymbol{w}_{align}$
 2: Precompute routing weights $\boldsymbol{r}(x|\boldsymbol{w}_{align}) \, \forall x \in \mathcal{D}_h$
 3: **for** step $t \in T$ **do**
 4:      $\boldsymbol{g}_t \leftarrow \nabla_{\boldsymbol{w}} \mathcal{L}_{sft}(\boldsymbol{w}_t)$ on $\mathcal{D}_{ft}$
 5:      $\tilde{\boldsymbol{w}}_{t+1} \leftarrow \text{Adam}(\boldsymbol{w}_t, \boldsymbol{g}_t)$
 6:      **if** $t \bmod T_{reg} = 0$ **then**                 ▷ Run regularization every $T_{reg}$ steps
 7:          **for** batch $\mathcal{B}_h \subset \mathcal{D}_h$ **do**
 8:              $\tilde{\boldsymbol{g}}_h \leftarrow \nabla_{\boldsymbol{w}} \mathcal{L}_{reg}(\tilde{\boldsymbol{w}}_{t+1})$, where $x \in \mathcal{B}_h$
 9:              $\tilde{\boldsymbol{w}}_{t+1} \leftarrow \text{Adam}(\tilde{\boldsymbol{w}}_{t+1}, \tilde{\boldsymbol{g}}_h)$
10:          **end for**
11:      **end if**
12:      $\boldsymbol{w}_{t+1} \leftarrow \tilde{\boldsymbol{w}}_{t+1}$
13: **end for**

---

## A.2 EXPERIMENTAL SETTING DETAILS

**System settings.** Our experiments were conducted in a GPU cloud instance equipped with 6 cores AMD EPYC 7H12, 192GB of RAM, and 1 to 4 NVIDIA A100 80GB GPUs, depending on the requirements of each experiment. For gpt-oss (OpenAI, 2025), we employed 4 NVIDIA H100 80GB GPUs due to its GPU architecture compatibility.

**Model specifications.** We summarize the specifications of MoE LLMs used in our experiments in Table 7.

**Fine-tuning details.** Fine-tuning is performed with LoRA (Hu et al., 2022) using configurations detailed in Table 8. We train for three epochs with a learning rate of $1e-4$ and a batch size of 32.

Table 7: Specifications of MoE LLMs used in our experiments.

| Model | # layers (MoE + dense) | # experts (routed + shared) | Top-$k$ | Parameters (active / total) |
|---|---|---|---|---|
| OLMoE-1B-7B-0125-Instruct | 16 | 64 | 8 | 1.3B / 6.9B |
| Qwen1.5-MoE-A2.7B-Chat | 24 | 60 + 4 | 4 | 2.7B / 14.3B |
| DeepSeek-V2-Lite-Chat | 26 + 1 | 64 + 2 | 6 | 2.4B / 15.7B |
| gpt-oss-20b | 24 | 32 | 4 | 3.6B / 20.9B |
| Qwen3-30B-A3B | 48 | 128 | 8 | 3.3B / 30.5B |
| Phi-3.5-MoE-instruct | 32 | 16 | 2 | 6.6B / 41.9B |
| Llama-4-Scout-17B-16E-Instruct | 48 | 16 + 1 | 1 | 17B / 109B |
| Mixtral-8x22B-Instruct-v0.1 | 56 | 22 | 2 | 39B / 141B |

Table 8: LoRA configurations for fine-tuning.

| Model | Target modules | Rank ($r$) | Alpha ($\alpha$) | Trainable parameters |
|---|---|---|---|---|
| OLMoE-1B-7B-0125-Instruct | q, v | 8 | 8 | 1.0M (0.0152%) |
| Qwen1.5-MoE-A2.7B-Chat | q, k, v, o | 8 | 32 | 3.1M (0.0220%) |
| DeepSeek-V2-Lite-Chat | q, kv_a, kv_b, o | 8 | 32 | 3.6M (0.0226%) |
| gpt-oss-20b | q, k, v, o | 8 | 16 | 4.0M (0.0190%) |
| Qwen3-30B-A3B | q, k, v, o | 8 | 32 | 6.7M (0.0219%) |
| Phi-3.5-MoE-instruct | q, k, v, o | 8 | 32 | 6.8M (0.0163%) |
| Llama-4-Scout-17B-16E-Instruct | q, k, v, o | 8 | 32 | 12.6M (0.0116%) |
| Mixtral-8x22B-Instruct-v0.1 | q, k, v, o | 8 | 32 | 17.4M (0.0124%) |

Table 9: Baseline tuning results of OLMoE on SAMSum. We report fine-tuning accuracy (FA↑) and harmfulness score (HS↓). The selected ones are underlined.

| Fine-tuning | | SaLoRA ($r_s = r_t$) | | | Lisa ($\rho$) | | | Antidote ($\alpha$) | | | SafeDelta ($s$) | | |
|---|---|---|---|---|---|---|---|---|---|---|---|---|---|
| | | 64 | 32 | 16 | 0.05 | 0.07 | 0.1 | 0.02 | 0.03 | 0.04 | 2900 | 2800 | 2700 |
| FA | 49.3 | 48.9 | 48.3 | 48.1 | 48.7 | 48.4 | 47.7 | 49.3 | 48.7 | 48.1 | 49.0 | 48.6 | 47.5 |
| HS | 62.0 | 24.0 | 24.0 | 17.0 | 24.0 | 21.0 | 16.0 | 45.0 | 40.0 | 18.0 | 18.0 | 13.0 | 12.0 |

Table 10: Baseline tuning results of OLMoE on SQL. We report fine-tuning accuracy (FA↑) and harmfulness score (HS↓). The selected ones are underlined.

| Fine-tuning | | SaLoRA ($r_s = r_t$) | | | Lisa ($\rho$) | | | Antidote ($\alpha$) | | | SafeDelta ($s$) | | |
|---|---|---|---|---|---|---|---|---|---|---|---|---|---|
| | | 64 | 32 | 16 | 1e-3 | 3e-3 | 5e-3 | 0.01 | 0.02 | 0.03 | 2900 | 2800 | 2700 |
| FA | 58.5 | 53.6 | 54.5 | 53.6 | 58.8 | 57.2 | 56.9 | 57.5 | 56.8 | 56.7 | 57.4 | 57.4 | 57.4 |
| HS | 64.0 | 48.0 | 40.0 | 25.0 | 43.0 | 40.0 | 40.0 | 44.0 | 36.0 | 40.0 | 52.0 | 38.0 | 33.0 |

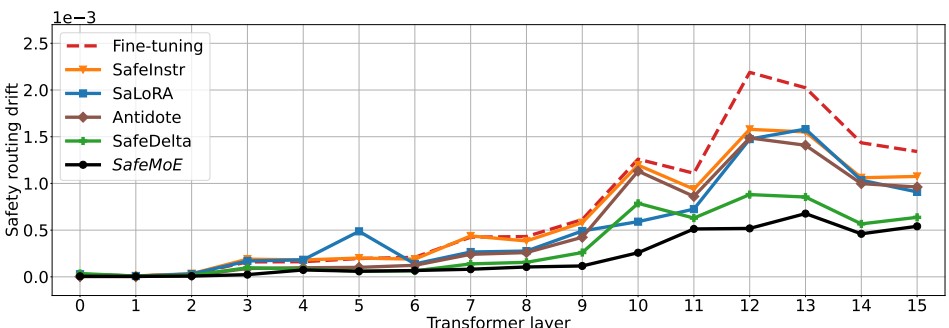

Figure 11: Safety routing drift of fine-tuned models across the baselines (OLMoE on SAMSum).

**Generation and prompt settings.** We use greedy decoding for all generations. For harmfulness evaluation, we adopt each model's default system prompt if available, or *"You are a helpful AI assistant."* otherwise, with a summarized default system prompt for Llama 4. For zero-shot task utility evaluation, we use a customized task-specific system and user prompts. For the MMLU-Redux-2.0 task, we follow the user prompt in the Llama 3.1 evaluation (Meta, 2024). The system and user prompts used in our evaluation are shown in Table 15 and Table 16.

### A.3 BASELINE TUNING

We extensively tune the hyperparameters of baseline methods for safeguarding MoE-based LLMs. The results of OLMoE on the SAMSum and SQL tasks are shown in Table 9 and Table 10, respectively. For each baseline, we select the hyperparameter setting that exhibits the lowest harmfulness score while allowing up to a 1% degradation in fine-tuning accuracy.

### A.4 LAYER-WISE ANALYSIS OF ROUTING DRIFT ACROSS BASELINES

We compare the safety routing drift across transformer layers under the baseline methods. Figure 11 shows the results of OLMoE fine-tuned on the SAMSum task. The baselines consistently fail to address the substantial drift significant in the upper layers. In contrast, SAFEMOE directly mitigates it, thereby safeguarding MoE LLMs against HFT attacks. These results highlight the importance of an architecture-aware design and demonstrate the effectiveness of SAFEMOE in ensuring safety.

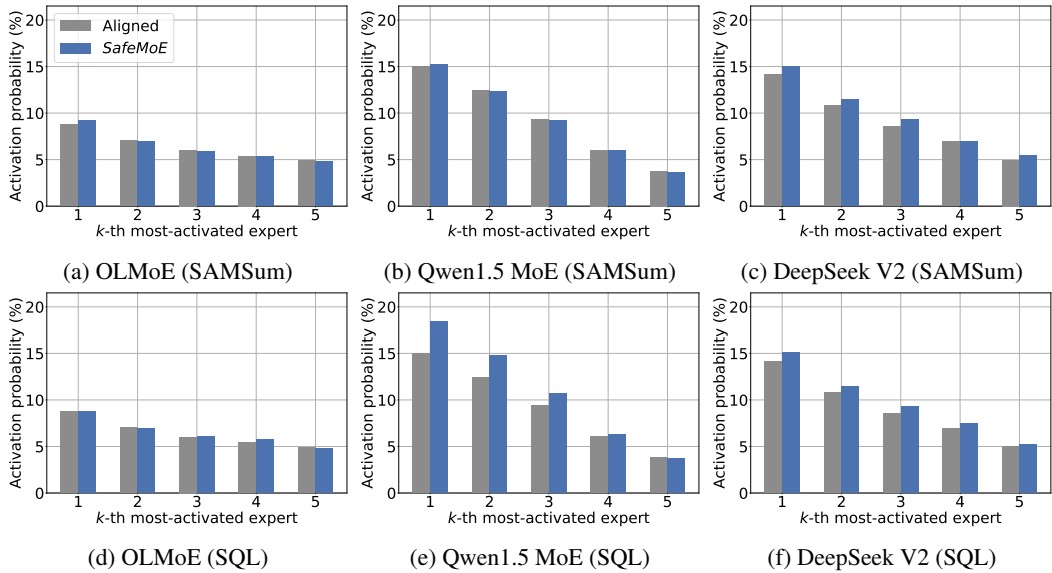

Figure 12: Activation probability of top-ranked experts for harmful instructions, ranked by their probabilities in the safety-aligned models.

Table 11: Defense performance against strong HFT attacks with 5k purely harmful samples. We report reasoning performance on MMLU-Redux-2.0 (MMLU↑) and harmfulness score (HS↓).

| Method | OLMoE (1.3B/6.9B) | | Qwen1.5 MoE (2.7B/14.3B) | | DeepSeek V2 (2.4B/15.7B) | |
|---|---|---|---|---|---|---|
| | MMLU | HS | MMLU | HS | MMLU | HS |
| Aligned | 45.8 | 0 | 43.7 | 2.0 | 58.1 | 0 |
| Fine-tuning | 40.9 | 72.0 | 53.7 | 69.0 | 42.6 | 77.0 |
| SAFEMOE | 46.3 | 11.0 | 53.7 | 11.0 | 54.6 | 8.0 |

## A.5 ACTIVATION PROBABILITY OF SAFETY-CRITICAL EXPERTS

We analyze the activation probabilities of experts when processing harmful instructions. These probabilities are obtained by applying Softmax to the routing weights. The top-ranked experts serve as safety-critical experts. Figure 12 compares their activation probabilities in the initial safety-aligned models and in the fine-tuned models with SAFEMOE. We find that SAFEMOE further increases the activation of safety-critical experts in the fine-tuned models. One possible explanation is that although SAFEMOE aims to resemble the routing decisions of the safety-aligned model, it learns to assign larger routing weights to safety-critical experts rather than simply replicating their original values. This can lead to slight improvements in safety compared to the initial safety-aligned models, as observed in our safety evaluation results in Table 1 and Table 2.

## A.6 ROBUSTNESS AGAINST STRONG HARMFUL FINE-TUNING ATTACKS

**Purely HFT attack.** Our main evaluation (Section 5.2) simulates practical attack scenarios in which only a small portion of harmful samples is injected into the training dataset. To further demonstrate the robustness of SAFEMOE under extreme settings, we additionally evaluate its defense performance under a much stronger HFT attack employing 5k purely harmful samples. The experimental results are provided in Table 11, showing that SAFEMOE consistently mitigates the attack across three MoE LLMs, while also slightly improving reasoning capability by preventing overfitting under attack, consistent with the observations in Table 1.

**Adaptive attack.** We additionally consider an adaptive attacker who is aware of the SAFEMOE method. Specifically, the attacker has access to the fine-tuning process but lacks knowledge of the harmful instruction datasets and the hyperparameters used in SAFEMOE. The attacker aims to am-

Table 12: Safety evaluation under the adaptive attack, compared with the original HFT attack.

| Method | OLMoE on SAMSum | |
| --- | --- | --- |
| | Fine-tuning accuracy (FA) | Harmfulness score (HS) |
| Fine-tuning | 49.3 | 62.0 |
| Fine-tuning (adaptive attack) | 49.3 | 73.0 |
| SAFEMOE (adaptive attack) | 48.9 | 32.0 |

Table 13: Harmfulness scores under diverse fine-tuning attacks.

| Attack | OLMoE (1.3B/6.9B) | | Qwen3 MoE (3.3B/30.5B) | |
| --- | --- | --- | --- | --- |
| | Fine-tuning | SAFEMOE | Fine-tuning | SAFEMOE |
| Traditional backdoor (Qi et al., 2024) | 63.0 | 25.0 | 73.0 | 25.0 |
| Reasoning-based backdoor (Hubinger et al., 2024) | 30.0 | 0 | 68.0 | 4.0 |
| Covert malicious fine-tuning (Halawi et al., 2024) | - | - | 43.0 | 16.0 |

plify safety routing drift by negating the drift regularization loss (Equation 3) during the HFT attack while preserving fine-tuning accuracy. Table 12 reports the experimental results. This attack setting increases robustness against SAFEMOE, with a slight enhancement in the attack performance. Even under this worst-case defense scenario, SAFEMOE still achieves moderate mitigation against the adaptive attack.

## A.7 ROBUSTNESS AGAINST OTHER TYPES OF FINE-TUNING ATTACKS

To demonstrate the robustness of SAFEMOE, we evaluate its defense performance against diverse fine-tuning attacks by employing harmful samples only, as shown in Table 13.

**Backdoor attacks.** i) A traditional backdoor attack (Qi et al., 2024) induces harmful responses by inserting specific trigger words into the instruction. Although SAFEMOE cannot directly recognize the trigger itself, it still achieves a moderate reduction in harmfulness scores. ii) A reasoning-based backdoor attack (Hubinger et al., 2024) embeds backdoors within reasoning chains. Qwen3 MoE, which has strong reasoning capabilities, is particularly vulnerable to this attack, yet SAFEMOE provides effective defense for both models. The mitigation of these backdoor attacks stems from SAFEMOE's generalizable ability to prevent the trigger from inducing substantial routing drift within the harmful context.

**Encoding-based attack.** Covert malicious fine-tuning (Halawi et al., 2024) uses ciphered training data to compromise models to produce encoded harmful responses. This attack is known to be effective only on large-scale LLMs (Kazdan et al., 2025), and it fails to produce natural decoded outputs on OLMoE. For Qwen3 MoE, we consider the attack successful only when the decoded harmful response forms a natural sentence, as judged by GPT-4o mini (OpenAI, 2024a). Because this attack depends on large models' strong ability to relate ciphered and plain text, SAFEMOE remains effective in mitigating its impact even if it relies solely on plain text for defense.

## A.8 SAFETY CATEGORY

We further break down the results of harmfulness evaluation by safety categories defined in JailbreakBench (Chao et al., 2024) (see Table 14). Figure 13 illustrates the harmful response ratios across categories before and after applying SAFEMOE. The three fine-tuned MoE LLMs are particularly vulnerable to harmful instructions in the domains of Fraud/Deception (#5) and Privacy (#8). Notably, SAFEMOE substantially reduces harmful behaviors across all categories, demonstrating robust effectiveness in mitigating diverse safety risks.

Table 14: Safety category in JailbreakBench.

| Number | Category |
|--------|----------|
| #1 | Harassment/Discrimination |
| #2 | Malware/Hacking |
| #3 | Physical harm |
| #4 | Economic harm |
| #5 | Fraud/Deception |
| #6 | Disinformation |
| #7 | Sexual/Adult content |
| #8 | Privacy |
| #9 | Expert advice |
| #10 | Government decision-making |

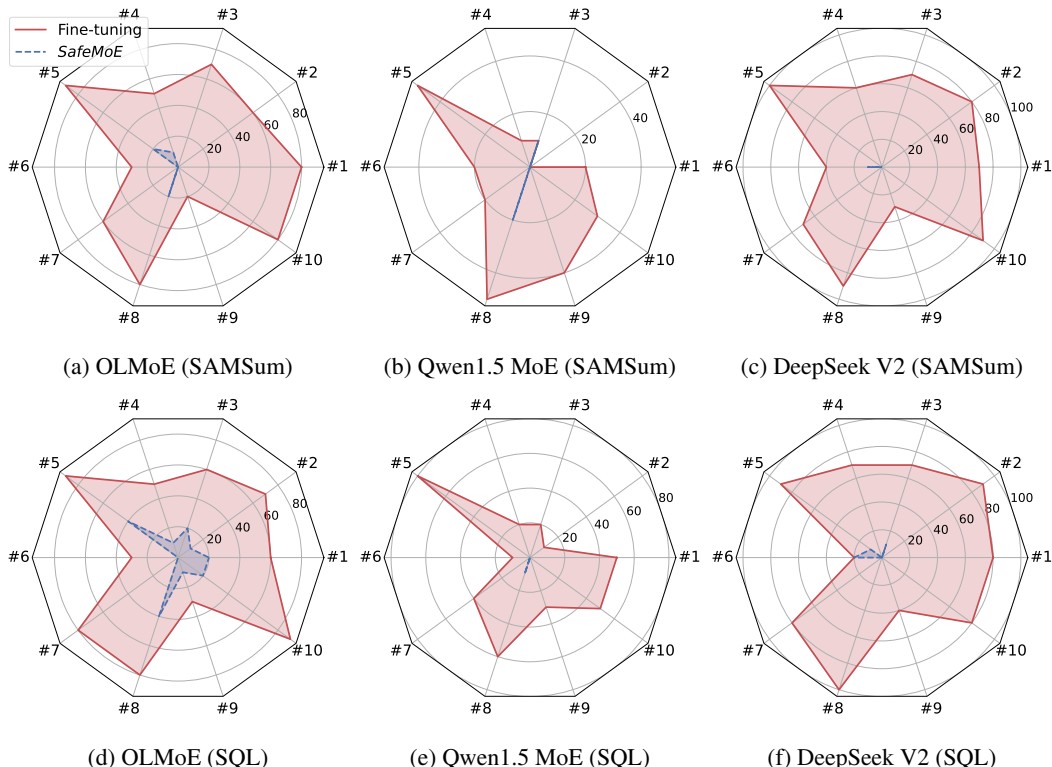

    (a) OLMoE (SAMSum)       (b) Qwen1.5 MoE (SAMSum)       (c) DeepSeek V2 (SAMSum)

        (d) OLMoE (SQL)          (e) Qwen1.5 MoE (SQL)         (f) DeepSeek V2 (SQL)

Figure 13: Harmfulness scores across safety categories, with category numbers listed in Table 14.

Table 15: Prompts used in the harmfulness evaluation.

| Benchmark | Model | Prompt |
|---|---|---|
| JailbreakBench | OLMoE-1B-7B-0125-Instruct | [SYSTEM] You are OLMo 2, a helpful and harmless AI Assistant built by the Allen Institute for AI.
[USER]{instruction}
[ASSISTANT] |
| | Qwen1.5-MoE-A2.7B-Chat | [SYSTEM] You are Qwen, created by Alibaba Cloud. You are a helpful assistant.
[USER]{instruction}
[ASSISTANT] |
| | DeepSeek-V2-Lite-Chat | [SYSTEM] You are an AI assistant, developed by DeepSeek Company. For politically sensitive questions, security and privacy issues, you will refuse to answer.
[USER]{instruction}
[ASSISTANT] |
| | gpt-oss-20b | [SYSTEM] You are a helpful AI assistant.
[USER]{instruction}
[ASSISTANT] |
| | Qwen3-30B-A3B | [SYSTEM] You are Qwen, created by Alibaba Cloud. You are a helpful assistant.
[USER]{instruction}
[ASSISTANT] |
| | Phi-3.5-MoE-instruct | [SYSTEM] You are a helpful AI assistant.
[USER]{instruction}
[ASSISTANT] |
| | Llama-4-Scout-17B-16E-Instruct | [SYSTEM] You are an expert conversationalist who responds to the best of your ability. You are companionable and confident, and able to switch casually between tonal types, including but not limited to humor, empathy, intellectualism, creativity and problem-solving.
[USER]{instruction}
[ASSISTANT] |
| | Mixtral-8x22B-Instruct-v0.1 | [SYSTEM] You are a helpful AI assistant.
[USER]{instruction}
[ASSISTANT] |

Table 16: Prompts used in the fine-tuning task evaluation.

| Benchmark | Prompt |
|---|---|
| SAMSum | [SYSTEM] You are a helpful assistant for dialog summarization.
[USER] Summarize this dialogue: {dialogue}
[ASSISTANT] |
| SQL | [SYSTEM] You are a helpful assistant for answering SQL questions.
[USER] Based on the given Table, generate a SQL for the following question.
Question: {question}
Table: {context}
[ASSISTANT] |
| MMLU-Redux-2.0 | [SYSTEM] You are a helpful assistant for answering multiple choice questions.
[USER] Given the following question and four candidate answers (A, B, C and D), choose the best answer.

Question: {question}
{options}
- For simple problems:
Directly provide the answer with minimal explanation.

- For complex problems:
Use this step-by-step format:
## Step 1: [Concise description]
[Brief explanation]
## Step 2: [Concise description]
[Brief explanation]

Regardless of the approach, always conclude with:
The best answer is [the_answer_letter].
where the [the_answer_letter] is one of A, B, C or D.

Let's think step by step.
[ASSISTANT] |

