# OpenReview forum: "SafeMoE: Safe Fine-Tuning for MoE LLMs by Aligning Harmful Input Routing"
_ICLR.cc/2026/Conference — ICLR 2026 Poster_

### Official Review · Reviewer_Maxq · 2025-10-16

**Soundness:** 3
**Presentation:** 3
**Contribution:** 2
**Rating:** 6
**Confidence:** 4

**Summary:**

This manuscript proposes SAFEMOE, a safe fine-tuning method tailored to Mixture-of-Experts (MoE) large language models (LLMs) to address the vulnerability of "safety routing drift"—a phenomenon where fine-tuning (both harmful and benign) causes the model’s routing decisions for harmful inputs to deviate from those of the initial safety-aligned model, thereby deactivating safety-critical experts and undermining safety. SAFEMOE mitigates this drift by introducing a regularization term that minimizes the KL-divergence between the routing weight distributions of the fine-tuned model and the safety-aligned model for harmful inputs. Extensive experiments on 8 MoE LLMs (ranging from 7B to 141B parameters, e.g., OLMoE, gpt-oss, Llama 4) demonstrate that SAFEMOE effectively reduces harmfulness scores (e.g., from 62.0 to 5.0 for OLMoE on the SAMSum task) while preserving task utility (≤1% degradation) and incurring only ~2% training overhead. The work also verifies the strong correlation between safety routing drift and model harmfulness, highlighting the importance of architecture-aware safety designs for MoE LLMs.

**Strengths:**

- Architecture-Aware Targeting: Unlike existing defenses (e.g., SafeInstr, SaLoRA) designed for monolithic LLMs, SAFEMOE directly addresses the unique vulnerability of MoE LLMs—their reliance on dynamic routing to safety-critical experts. This design fills a critical gap in current LLM safety research, where MoE-specific security challenges have been largely overlooked.
- Strong Empirical Validation: The authors conduct comprehensive experiments across diverse MoE LLMs (varying in parameter size, expert configurations, and vendors) and tasks (dialogue summarization, SQL generation). The consistent effectiveness of SAFEMOE across models (e.g., reducing harmfulness for 141B-parameter Mixtral) and benchmarks (JailbreakBench, HEx-PHI) enhances the work’s reliability.
- Practical Trade-Off Balance: SAFEMOE achieves a favorable balance between safety, utility, and efficiency. It reduces harmfulness significantly without sacrificing task performance (≤1% accuracy loss) and maintains low overhead (~2%), making it suitable for real-world fine-tuning services (e.g., OpenAI/Google API platforms).
- Clear Mechanistic Insight: The introduction and validation of "safety routing drift" as a key metric (via KL-divergence) provides a clear mechanistic explanation for MoE LLM safety degradation during fine-tuning. This insight not only supports SAFEMOE’s design but also guides future research on MoE security.

**Weaknesses:**

- While SAFEMOE addresses a practical gap, its core technical components (KL-divergence regularization, bi-level greedy optimization) are not fundamentally new—they adapt existing regularization and optimization paradigms (as cited in the paper) to MoE LLMs. Thus, the work does not introduce groundbreaking methodological innovations, limiting its potential to be a "landmark" contribution in LLM safety.
- The study’s threat model implicitly assumes users are either "benign" (conducting standard task fine-tuning) or "moderately malicious" (injecting a small fraction of harmful samples, e.g., 500 out of 5.5k samples in HFT). It does not rigorously evaluate scenarios where attackers use purely malicious datasets (100% harmful instructions) to fine-tune the model, leaving uncertainty about SAFEMOE’s robustness against extreme adversarial intent.
- Multiple figures (e.g., Figure 1b, 5b) show that safety routing drift and harmfulness scores increase disproportionately rapidly in the early stages of fine-tuning (e.g., first 100 steps for OLMoE). The manuscript does not provide a detailed mechanistic explanation for this phenomenon (e.g., whether it stems from rapid updates to gating network parameters or skewed gradient flow from task/harmful samples), weakening the depth of its analysis.

**Questions:**

Please refer to the "Weaknesses" Section.

---

> ### Author Response · Authors · 2025-11-21
> **Rebuttal by Authors**
>
> We sincerely appreciate the reviewer’s constructive feedback, particularly for comments regarding our paper’s contributions and the need for a deeper analysis of fine-tuning mechanisms. We have uploaded the revised manuscript that incorporates these responses and results, with corresponding updates specified for each question. We hope these clarifications address the concerns raised. If you have any further questions or comments, we would be grateful to receive them.
>
> ### **1. Contributions of the paper**
>
> The main contribution of our work lies in uncovering previously unseen safety vulnerabilities inherent to the MoE architecture through an in-depth analysis of its unique safety mechanisms, and in designing defense techniques that are immediately applicable to existing MoE-based LLMs. We believe that our practical analysis and findings represent a significant step forward in improvingLLM safety.
>
> ### **2. HFT attack using purely harmful data**
>
> > Revision in *Section A.6*, *Table 11*
>
> While our main evaluation simulates practical attack scenarios relying on only a small portion of harmful samples, the experiments in Table 2 (Section 5.2) target extreme attack settings that use purely harmful datasets. SafeMoE effectively protects large-scale MoE LLMs using only a small number (100) of harmful instructions. To further demonstrate the robustness of SafeMoE, we additionally evaluated its defense performance under an even stronger HFT attack that employs 5k purely harmful samples. The experimental results are provided below, showing that SafeMoE consistently mitigates the attack across three MoE LLMs, while also slightly improving MMLU-Redux-2.0 performance by preventing overfitting under attack, consistent with the observations in Table 2.
>
> | Method      | OLMoE                |                   | Qwen1.5 MoE          |                   | DeeSeek V2           |                   |
> |-------------|----------------------|-------------------|----------------------|-------------------|----------------------|-------------------|
> |             | Fine-tuning accuracy | Harmfulness score | Fine-tuning accuracy | Harmfulness score | Fine-tuning accuracy | Harmfulness score |
> | Aligned     | 45.8                 | 0                 | 43.7                 | 2.0               | 58.1                 | 0                 |
> | Fine-tuning | 40.9                 | 72.0              | 53.7                 | 69.0              | 42.6                 | 77.0              |
> | SafeMoE     | 46.3                 | 11.0              | 53.7                 | 11.0              | 54.6                 | 8.0               |
>
> ### **3. Mechanistic explanation for the early find-tuning stages**
>
> As a deeper analysis of the training dynamics, we tracked both the training losses and routing drifts for task samples and harmful samples. The results show that the losses for both types of samples drop rapidly during the early stages and converge stably after 200 steps (the first epoch corresponds to 172 steps), consistent with a concurrent increase in routing drift. Notably, despite harmful samples constituting only a small fraction of the training data (~10%), the safety of MoE LLMs is quickly compromised within just a few training steps, aligning with prior observations [R1, R2].
>
> | Training step | Training loss |         | Routing drift |         |
> |---------------|---------------|---------|---------------|---------|
> |               | Task (SAMSum) | Harmful | Task (SAMSum) | Harmful |
> | 50            | 1.97          | 2.62    | 1.7e-4        | 6.7e-4  |
> | 100           | 1.25          | 1.94    | 2.1e-4        | 8.8e-4  |
> | 150           | 1.2           | 1.83    | 2.5e-4        | 9.8e-4  |
> | 200           | 1.18          | 1.79    | 1.8e-4        | 1.0e-3  |
> | 250           | 1.19          | 1.84    | 1.8e-4        | 1.0e-3  |
> | 300           | 1.18          | 1.83    | 1.9e-4        | 1.0e-3  |
> | 350           | 1.18          | 1.82    | 1.6e-4        | 1.0e-3  |
> | 400           | 1.18          | 1.82    | 1.6e-4        | 1.0e-3  |
> | 450           | 1.18          | 1.82    | 1.6e-4        | 1.0e-3  |
> | 500           | 1.18          | 1.82    | 1.6e-4        | 1.0e-3  |
>
> [R1] Qi et al., “Fine-tuning Aligned Language Models Compromises Safety, Even When Users Do Not Intend To!”, ICLR 2024 \
> [R2] Souly et al., “Poisoning Attacks on LLMs Require a Near-constant Number of Poison Samples” arXiv 2024.

---

> ### Comment · Reviewer_Maxq · 2025-11-21
>
> Thank you for your thorough and thoughtful response to my comments. Most of the concerns I raised have been adequately addressed with clear explanations and necessary revisions. Based on your compelling rebuttal, I am pleased to revise my score upward.

---

> > ### Author Response · Authors · 2025-11-25
> >
> > We sincerely appreciate your response and the update. Your valuable feedback has been very helpful to strength the paper.

---

### Official Review · Reviewer_6cTn · 2025-10-17

**Soundness:** 4
**Presentation:** 4
**Contribution:** 3
**Rating:** 6
**Confidence:** 5

**Summary:**

This paper study harmful fine-tuning defense on MOE LLMs. As a solution, this paper propose a simple regularizer to mitigate the drift of the routing weights.

**Strengths:**

1. This is the first paper that systematically study harmful fine-tuning on MOE models.
2. This paper discovers an important finding that harmful fine-tuning makes the routing  weights drift away from that of the aligned model.
3. The proposed method is simple and should work in my understanding.
4. Paper is extremely well written. The idea is presented clear with empirical evidence as motivations. The main experiments are also very comprehensive, justifying the effectiveness of the proposed method.

**Weaknesses:**

1. **A baseline should be added to comparison given their similarity.**
I suggest the authors to add a comparison with Lisa [1]. Similar to SAFEMOE,  Lisa adopts a proximal term which also somehow helps mitigate the harmful drift (although not only for routing weights). In addition, Lisa adds a few safety data in the fine-tuning process, which probably can help improve the mitigation effect. Therefore, I am wondering whether SAFEMOE can outperform Lisa even though Lisa has additional safety alignment data in the fine-tuning process.

2. **Add a few discussions  and do some exploration on expert-wise regularization.** I first need to mention a related MoE drift control method. Please check Section 5.1 in H3fusion[2].  They shows that by **adding more regularize intensity** to safety MOE expert, the drift of this safety MOE expert during fine-tuning will be smaller. However, they show that this regularizer is not beneficial, because large regularization intensity of safety expert will make the model **becomes even more unsafe** (See the stage 4 in Figure 1). This conclusion is **contradictory** to yours. The reason of this discrepancy is probably that they are adding regularizer to only the safety expert but you are adding uniform regularization to all the experts together.  Combining the two papers, I think the a potentially better solution might be: add the proximal regularization to the non-safety expert but don't add proximal regularization (or add negative proximal regularization) to the safety expert. I conjecture the gate drift will be uniformly become larger for all the experts (but not in the case like Figure 4 that you display), and constraining that of the safety expert will not help.  My conjecture might be or might not be wrong. Could you perform this experiment to verify this alternative idea?



[1] Lisa: Lazy Safety Alignment for Large Language Models against Harmful Fine-tuning Attack

[2] H3Fusion: Helpful, Harmless, Honest Fusion of Aligned LLMs

3. Some related work are missing. I list out a few most recent work below and please consider to read and discuss them.




Unleashing the Unseen: Harnessing Benign Datasets for Jailbreaking Large Language Models

No, of course I can! Refusal Mechanisms Can Be Exploited Using Harmless Fine-Tuning Data

Harmful Fine-tuning Attacks and Defenses for Large Language Models: A Survey

Emergent Misalignment:Narrow finetuning can produce broadly misaligned LLMs

ESTIMATING WORST-CASE FRONTIER RISKS OF OPEN-WEIGHT LLMS

Your Agent May Misevolve: Emergent Risks in Self-evolving LLM Agents

Deep Ignorance: Filtering Pretraining Data Builds Tamper-Resistant Safeguards into Open-Weight LLMs

Tamper-Resistant Safeguards for Open-Weight LLMs

CTRAP: Embedding Collapse Trap to Safeguard Large Language Models from Harmful Fine-Tuning

Self-Destructive Language Model

Targeted Vaccine: Safety Alignment for Large Language Models against Harmful Fine-Tuning via Layer-wise Perturbation

Vulnerability-Aware Alignment: Mitigating Uneven Forgetting in Harmful Fine-Tuning

LoX: Low-Rank Extrapolation Robustifies LLM Safety Against Fine-tuning

Covert malicious finetuning: Challenges in safeguarding llm adaptation

Shape it Up! Restoring LLM Safety during Finetuning

AsFT: Anchoring Safety During LLM Fine-Tuning Within Narrow Safety Basin

 Gradient Surgery for Safe LLM Fine-Tuning

Panacea: Mitigating Harmful Fine-tuning for Large Language Models via Post-fine-tuning Perturbation

Refusal-Feature-guided Teacher for Safe Finetuning via Data Filtering and Alignment Distillation

When Style Breaks Safety: Defending Language Models Against Superficial Style Alignment

Anchoring Refusal Direction: Mitigating Safety Risks in Tuning via Projection Constraint

Why LLM Safety Guardrails Collapse After Fine-tuning: A Similarity Analysis Between Alignment and Fine-tuning Datasets

Toward Secure Tuning: Mitigating Security Risks from Instruction Fine-Tuning

Safety Subspaces are Not Distinct: A Fine-Tuning Case Study

MoGUV 2: Toward a Higher Pareto Frontier Between Model Usability and Security

Layer-Aware Representation Filtering: Purifying Finetuning Data to Preserve LLM Safety Alignment

Virus: Harmful Fine-tuning Attack for Large Language Models Bypassing Guardrail Moderation

Finetuning-Activated Backdoors in LLMs

 Safety alignment should be made more than just a few tokens deep

Mitigating Fine-tuning Risks in LLMs via Safety-Aware Probing Optimization

Fundamental Safety-Capability Trade-offs in Fine-tuning Large Language Models

Navigating the safety landscape: Measuring risks in finetuning large language models

Scaling Trends for Data Poisoning in LLMs

Towards Resilient Safety-driven Unlearning for Diffusion Models against Downstream Fine-tuning

Fundamental Safety-Capability Trade-offs in Fine-tuning Large Language Models

Detecting Adversarial Fine-tuning with Auditing Agents

Your Task May Vary: A Systematic Understanding of Alignment and Safety Degradation when Fine-tuning LLMs

There might be more exciting papers that are worth to read in this field and I encourage the authors to read more related papers and keep update on the field.

**Questions:**

I don't have further questions as I am kind of familiar with both harmful fine-tuning and MoE. This paper excites me, inspire me thinking and I do wish to get more insights from you.  I will consider to increase my score if the authors are willing to explore more on the direction I point out (see weakness 1&2), but otherwise I am also happy to keep my score.

---

> ### Author Response · Authors · 2025-11-21
> **Rebuttal by Authors**
>
> We sincerely appreciate the reviewer’s constructive feedback, particularly the suggestion to include an additional competitive baseline and to expand our discussion of related work such as H3Fusion [2]. We provide our detailed responses and additional experimental results below. We have uploaded the revised manuscript that incorporates these responses and results, with corresponding updates specified for each question. We hope these clarifications address the concerns raised. If you have any further questions or comments, we would be grateful to receive them.
>
> ### **1. Additional baseline (Lisa [1])**
>
> > Revision in *Section 5.1*, *Section 5.2*, *Table 9*, *Table 10*
>
> *1) SAMSum*
> | Method       | OLMoE |      | Qwen1.5 MoE |      | DeepSeek V2 |      |
> |--------------|-------|------|-------------|------|-------------|------|
> |              | Fine-tuning accuracy    | Harmfulness score   |  Fine-tuning accuracy           | Harmfulness score   |  Fine-tuning accuracy           | Harmfulness score   |
> | Fine-tuning  | 49.3  | 62.0 | 50.4        | 49.0 | 52.0        | 40.0 |
> | SafeMoE      | 48.9  | 5.0  | 50.6        | 0    | 51.0        | 1.0  |
> | Lisa         | 48.4  | 21.0 | 50.1        | 10.0 | 50.7        | 24.0 |
>
> *2) SQL*
> | Method       | OLMoE |      | Qwen1.5 MoE |      | DeepSeek V2 |      |
> |--------------|-------|------|-------------|------|-------------|------|
> |              | Fine-tuning accuracy    | Harmfulness score   | Fine-tuning accuracy          | Harmfulness score   | Fine-tuning accuracy          | Harmfulness score   |
> | Fine-tuning  | 58.5  | 64.0 | 70.2        | 37.0 | 70.1        | 72.0 |
> | SafeMoE      | 59.0  | 17.0 | 69.5        | 1.0  | 69.1        | 4.0  |
> | Lisa         | 57.2  | 40.0 | 68.5        | 13.0 | 68.9        | 22.0 |
>
> ### **2. Discussion on H3fusion [2]**
>
> We thank the reviewer for suggesting a new line of thought. However, we would like to clarify the distinction between the concept of drift in H3Fusion and that in our work. The *embedding drift* discussed in H3Fusion refers to variations in *the outputs of expert layers*, where a larger drift indicates higher expert capability (e.g., a larger embedding drift in the safety expert corresponds to stronger safety behavior). Therefore, regularizing the embedding drift of the safety expert within an MoE would weaken the overall safety, as demonstrated in H3Fusion.
>
> In contrast, the safety routing drift introduced in our work refers to changes in *routing weights*, which are produced by the gating network and determine expert activation. The contribution of each expert is governed by its corresponding value in the routing weights, which are dynamically computed for each input. Harmful fine-tuning causes these routing weights to deviate from the safe state of the initial aligned model. As a result, a larger safety routing drift makes the MoE model more harmful, and regularizing this drift effectively prevents safety degradation.
>
> Thus, the effects of these two types of drift are fundamentally different rather than contradictory. We would appreciate any further comments or discussion you may have.
>
> ### **3. Additional related work**
>
> > Revision in *Section1*, *Section 2* and *Section 6*
>
> We thank the reviewer for pointing out the missing related work. In the revised version, we have included several studies, particularly prior analyses of fine-tuning mechanisms (e.g., *Tamper-Resistant Safeguards for Open-Weight LLMs* and *Navigating the Safety Landscape: Measuring Risks in Finetuning Large Language Models*) and defense approaches that employ drift-regularization-like techniques (e.g., *Anchoring Refusal Direction: Mitigating Safety Risks in Tuning via Projection Constraint* and *Mitigating Fine-tuning Risks in LLMs via Safety-Aware Probing Optimization*). In addition, OpenAI’s analysis of fine-tuning risks for gpt-oss, a very recent MoE LLM, presented in *Estimating Worst-Case Frontier Risks of Open-Weight LLMs*, also has been included as a related study.

---

> ### Author Response · Authors · 2025-11-28
>
> Dear Reviewer
>
> We sincerely appreciate your constructive feedback once again. We have made substantial improvements based on your valuable suggestions. If possible, we would be grateful for any further comments. Please let us know if you feel that additional clarification or revisions are needed.
>
> Sincerely,\
> The authors

---

### Official Review · Reviewer_YmtJ · 2025-10-22

**Soundness:** 1
**Presentation:** 3
**Contribution:** 2
**Rating:** 2
**Confidence:** 4

**Summary:**

This paper proposes a defense method against harmful fine-tuning attacks on MoE LLMs. The core idea is to regularize the MoE routing decisions during fine-tuning.

**Strengths:**

* The proposed method is specially tailored for MoE LLMs according to their routing nature, which prior work has not considered.
* The experiments span across MoE LLMs of diverse sizes, and include important ablation studies on method hyperparameters.
* The paper writing flows well overall.

**Weaknesses:**

* Questionable motivation. The authors show that harmfulness score is highly correlated with safety routing drift. However, is harmfulness score also correlated with other weights (non-routing weights) of the MoE model? And if the method regularizes the drifts in other weights rather than the routing weights, will the harmfulness score also be suppressed? Without these ablations, the motivation to "preserve the initial routing decisions" is questionable.

  Accordingly, I think it's necessary to compare your method with the fine-tuning defense in [1], where the authors also propose to constrain the fine-tuned weights to the original weights.

* Lack of consideration of more complicated attacks and adaptive attacks. The experiments solely focus on  the attack setting of explicitly mixing "500 harmful samples from BeaverTails" in the fine-tuning dataset. However, there are many different HFT attacks nowadays. I would suggest the authors consider more complicated HFT attacks (e.g., [2]), and even backdoor HFT attacks [3,4]. Additionally, the authors should discuss potential adaptive attacks against their proposed defense.

  I'm particularly interested in seeing the results of backdoor HFT attacks, where a trigger is used for activating harmful LLM responses (the model would refuse harmful requests if a trigger is not present). I doubt whether this backdoor attack setting can break the proposed defense -- since SafeMoE can only preserve the routing decisions of plain harmful instructions (where trigger is not included), but cannot regularize the case when backdoor triggers are secretly injected into the harmful instructions.

* Lack of ablation on $T_{reg}$. According to Table 3, it seems the overhead of SafeMoE is quite low. So what if the method performs regularization steps more regularly? Can you show the tradeoff of overhead v.s. safety across diverse choices of $T_{reg}$?

* Lack of human evaluation of the reliability of Llama-Guard-4-12B.

[1] Safety Alignment Should Be Made More Than Just a Few Tokens Deep. ICLR 2025

[2] Covert Malicious Finetuning: Challenges in Safeguarding LLM Adaptation. ICML 2024

[3] Fine-tuning Aligned Language Models Compromises Safety, Even When Users Do Not Intend To! ICLR 2024

[4] Sleeper Agents: Training Deceptive LLMs that Persist Through Safety Training. Arxiv

**Questions:**

* In Figure 1, step 150, the safety routing drift of both SafeMoE and SaLoRA are similarly low (~0.7*1e-3), but their harmfulness scores are quite discrepant -- ~55% for SafeMoE and ~15% for SaLoRA. Why?
* In Figure 3, why use different benign datasets across the two settings (Alpaca v.s. SAMSum)?
* Can you include a comparison with directly freezing the router weights during fine-tuning (rather than regularizing them)?
* Does your method only work for MoE LLMs? Or can you adapt your method to work on non-MoE LLMs too?

---

> ### Author Response · Authors · 2025-11-21
> **Rebuttal by Authors (1/3)**
>
> We sincerely appreciate the reviewer’s constructive feedback, particularly the comments regarding the motivation and the additional attack scenarios that were not considered. We provide our detailed responses and additional experimental results below. We have uploaded the revised manuscript that incorporates these responses and results, with corresponding updates specified for each question. We hope these clarifications address the concerns raised. If you have any further questions or comments, we would be grateful to receive them.
>
> ### **1. Contribution of routing weights to safety**
> > Revision in *Section 3*, *Figure 4*, *Section 5.5*, *Table4*
>
> To further support our motivation and claims, we conducted two additional experiments that separately evaluate the influences of routing weights on MoE safety.
>
> *1) Inference with initial routing weights*
>
> We assess the harmfulness of fine-tuned models by conducting inference using the initial routing weights (i.e., those before fine-tuning) for each input. The results below show that this ablation setting also achieves substantial reductions in harmfulness, demonstrating the critical role of routing weights for MoE safety.
>
> | Fine-tuning step | HS (frozen routing weights) | HS (original fine-tuning) |
> |------------------|--------------------------------------------|---------------------------------|
> | 50               | 8.0                                        | 11.0                            |
> | 100              | 10.0                                       | 35.0                            |
> | 150              | 19.0                                       | 53.0                            |
> | 200              | 17.0                                       | 59.0                            |
> | 250              | 13.0                                       | 60.0                            |
> | 300              | 15.0                                       | 61.0                            |
> | 350              | 14.0                                       | 63.0                            |
> | 400              | 16.0                                       | 61.0                            |
> | 450              | 16.0                                       | 61.0                            |
> | 500              | 16.0                                       | 60.0                            |
> *HS: Harmfulness score
>
> *2) Blocking gradient flows from expert output vectors*
>
> We further conduct an ablation study on gradient flow to isolate and quantify the contribution of our routing-preservation approach. There are two forward paths (input → layer → output) in an MoE module: 1) *residual vector → gating network → routing weights*, which determines routing decisions, and 2) *residual vector → expert layers → expert output vectors*, which produces the expert representations.
>
> In our ablation, we block gradient flows from the expert output vectors back into the residual vector through the second path. This allows us to separately examine the effect of routing preservation without interference from residual vector updates induced by the previous transformer layers and the expert layers. The results below show that this ablation setting still achieves strong defense performance, supporting the validity of our claims.
>
> | Method             | Fine-tuning accuracy | Harmfulness score |
> |--------------------|----------------------|-------------------|
> | Fine-tuning        | 49.3                 | 62.0              |
> | SafeMoE            | 48.9                 | 5.0               |
> | SafeMoE (ablation) | 48.2                 | 18.0              |
>
> We believe these deeper analyses reinforce our understanding of MoE safety mechanisms and strengthen our motivation to preserve the initial routing weights.

---

> ### Author Response · Authors · 2025-11-21
> **Rebuttal by Authors (2/3)**
>
> ### **2. Defense against HFT attack variants**
>
> > Revision in *Section A.6*, *Table 12*, *Section A.7*, *Table 13*
>
> We examined the defense effectiveness of SafeMoE against fine-tuning attack variants [2,3,4] with the experimental results below. Because some of these attacks are less effective on OLMoE (7B), we additionally measured performance using the more capable Qwen3 MoE (30B).
>
> *1) Traditional backdoor [3] and reasoning-based [4] backdoor attack*
>
> SafeMoE achieves moderate defense performance against the traditional backdoor attack that relies on trigger words and strong defense performance against the backdoor attack exploiting reasoning chains. The mitigation of these backdoor attacks stems from SafeMoE’s generalizable ability to prevent the trigger from inducing substantial routing drift within the harmful context.
>
> | Attack                       | OLMoE (7B)       |                 | Qwen3 MoE (30B)  |                 |
> |------------------------------|------------------|-----------------|------------------|-----------------|
> |                              | HS (w/o SafeMoE) | HS (w/ SafeMoE) | HS (w/o SafeMoE) | HS (w/ SafeMoE) |
> | Traditional backdoor [3]     | 63.0             | 25.0            | 73.0             | 25.0            |
> | Reasoning-based backdoor [4] | 30.0             | 0               | 68.0             | 4.0             |
> *HS: Harmfulness score
>
> *2) Encoding-based attack [2]*
>
> Covert malicious fine-tuning [2] uses ciphered training data to compromise models to produce encoded harmful responses. This attack is only effective on large-scale models [R1], as mapping ciphered and plain text is challenging for typical LLMs such as OLMoE, which fails to produce natural decoded outputs. For Qwen3 MoE, we considered the attack successful only when the decoded harmful response forms a natural sentence, as judged by GPT-4o mini. Because this attack depends on large models’ strong ability to relate ciphered and plain text, SafeMoE remains effective in mitigating its impact even if it relies solely on plain text for defense.
>
> | Attack                       | OLMoE (7B)       |                 | Qwen3 MoE (30B)  |                 |
> |------------------------------|------------------|-----------------|------------------|-----------------|
> |                              | HS (w/o SafeMoE) | HS (w/ SafeMoE) | HS (w/o SafeMoE) | HS (w/ SafeMoE) |
> | Covert malicious fine-tuning [2]     | -             | -            | 43.0             | 16.0            |
> *HS: Harmfulness score
>
> *3) Adaptive attack*
>
> We further consider an adaptive attacker who is aware of SafeMoE and can access the fine-tuning process, but lacks knowledge of the harmful instruction datasets and the hyperparameters used by SafeMoE. The attacker attempts to increase safety routing drift by negating the drift regularization loss during the HFT attack while maintaining fine-tuning accuracy. Results for OLMoE on the SAMSum task (below) show that, even under this strong attack setting, SafeMoE still provides moderate mitigation.
>
> | Method                        | Fine-tuning accuracy | Harmfulness score |
> |-------------------------------|----------------------|-------------------|
> | Fine-tuning                   | 49.3                 | 62.0              |
> | Fine-tuning (adaptive attack) | 49.3                 | 73.0              |
> | SafeMoE (adaptive attack)     | 48.9                 | 32.0              |
>
> ### **3. Ablation on $T_{reg}$ and overhead trade-off**
>
> > Revision in *Section 5.6*, *Figure 9c*
>
> We designed the SafeMoE optimization to be highly scalable while maintaining sufficient effectiveness in our settings. To provide further insight into its implementation, we examined the trade-off between computational overhead and $T_{reg}$, as shown in the results below. As expected, a smaller $T_{reg}$ (i.e., more frequent regularization) introduces greater overhead but yields stronger defense. We argue that the resulting overhead remains reasonable and tolerable, when compared to our baseline methods in Section 5.4.
>
> | Regularization period $T_{reg}$ | Fine-tuning accuracy   | Harmfulness score   | Extra time (%) |
> |-------|------|------|----------------|
> | 100   | 48.5 | 3.0  | 23.44s (2.89%) |
> | 150   | 49.1 | 7.0  | 17.26s (2.13%) |
> | 200   | 49.3 | 10.0 | 14.14s (1.75%) |
> | 250   | 49.1 | 13.0 | 14.14s (1.75%) |
> | 300   | 49.1 | 20.0 | 11.09s (1.37%) |
>
> [R1] Kazdan et al., “No, of Course I Can! Deeper Fine-Tuning Attacks That Bypass Token-Level Safety Mechanisms”, arXiv 2025

---

> ### Author Response · Authors · 2025-11-21
> **Rebuttal by Authors (3/3)**
>
> ### **4. The reliability of Llama-Guard-4-12B**
>
> We use Llama-Guard-4-12B [R2], a model specialized for detecting and classifying harmful content, to quantitatively assess harmfulness. This model exhibits strong performance and substantially outperforms Llama-Guard-3-8B [R3]. Notably, even the earlier version already surpasses GPT-4-based judging [R3] and has been widely adopted in safety evaluation studies [R4, R5, R6]. Therefore, we believe that using Llama-Guard-4-12B does not undermine the validity of our evaluation. To address the reviewer’s concern, two authors manually inspected 100 randomly sampled prompt-response pairs across the three MoE LLMs in Table 1 evaluated by Llama-Guard-4-12B. We did not observe any misclassifications in these samples.
>
> ### **5. Similar safety routing drifts of SafeMoE and SafeLoRA in Figure 1**
>
> > Revision in *Figure 1a*, *Figure 6a*
>
> We thank the reviewer for pointing this out. After submission, we discovered that some safety routing drift values were mistakenly measured on $D_h$ instead of the evaluation benchmark. We have corrected these results and thoroughly verified that no issues are present in the other evaluations. We sincerely apologize for the confusion. In the corrected results, the gap between SafeMoE and SafeLoRA is more reasonable and aligns with the observed changes in harmfulness scores across training steps.
>
> ### **6. Two types of benign datasets**
>
> The Alpaca dataset is a widely used benign dataset known to unintentionally degrade the safety of LLMs [R7, R8, R9, R10]. For consistency with prior studies, we evaluate MoE LLMs on this dataset for benign fine-tuning cases. For the HFT attacks in our main evaluation (Section 5), we employed a task-specific dataset (SAMSum) to concretely assess task-performance trade-offs, as Alpaca consists of general-purpose instructions. The HFT attack results reported in Figure 3 (Section 3) correspond to these main evaluation settings.
>
> ### **7. Adaptation to non-MoE LLMs**
> While many prior studies have proposed effective defenses for non-MoE LLMs, our work introduces a highly scalable and readily applicable defense that leverages the unique mechanism of the MoE architecture. The current design of our defense is intentionally specialized for MoE LLMs, yet enabling it to achieve superior effectiveness on MoE LLMs. Therefore, with respect, we would appreciate further clarification on the adaptation to non-MoE LLMs that the reviewer has in mind, so we can address this comment more carefully.
>
> [R2] https://huggingface.co/meta-llama/Llama-Guard-4-12B \
> [R3] https://huggingface.co/meta-llama/Llama-Guard-3-8B \
> [R4] Chen et al., “Red Teaming GPT-4V: Are GPT-4V Safe Against Uni/Multi-Modal Jailbreak Attacks?”, ICLR (Workshop) 2024 \
> [R5] Hammoud et al., “Model Merging and Safety Alignment: One Bad Model Spoils the Bunch”, EMNLP (Findings) 2024 \
> [R6] Zheng et al., “Improved Few-Shot Jailbreaking Can Circumvent Aligned Language Models and Their Defenses”, NeurlPS 2024 \
> [R7] Qi et al., “Fine-tuning Aligned Language Models Compromises Safety, Even When Users Do Not Intend To!”, ICLR 2024 \
> [R8] Bhardwaj et al., ”Language Models are Homer Simpson! Safety Re-Alignment of Fine-tuned Language Models through Task Arithmetic”, ACL 2024 \
> [R9] Lu et al., “Safe Delta: Consistently Preserving Safety when Fine-Tuning LLMs on Diverse Datasets”, ICLR 2025 \
> [R10] Li et al., “SaLoRA: Safety-Alignment Preserved Low-Rank Adaptation” ICLR 2025

---

> ### Author Response · Authors · 2025-11-28
>
> Dear Reviewer
>
> We sincerely appreciate your constructive feedback once again. We have made substantial improvements based on your valuable suggestions. If possible, we would be grateful for any further comments. Please let us know if you feel that additional clarification or revisions are needed.
>
> Sincerely,\
> The authors

---

### Official Review · Reviewer_Ysqm · 2025-10-31

**Soundness:** 2
**Presentation:** 3
**Contribution:** 3
**Rating:** 4
**Confidence:** 3

**Summary:**

The authors consider the problem of preventing subsequent fine-tunes of an aligned MoE-based LLM from eliciting harmful behavior or capabilities. In particular, this setting assumes the attacker can choose the fine-tuning data mix, but otherwise has no access to the model parameters or the training setup. Such attacks are referred to as Harmful Fine-Tuning (HFT). This threat model is relevant for e.g. the OpenAI fine-tuning API, where (potentially malicious) users can fine-tune OpenAI models on their own data.

Prior work has identified the existence of safety-critical experts in aligned MoE models. The authors posit that one of the core mechanisms of HFT’s success in MoE models is routing drift, where the expert assignment changes substantially during fine-tuning, avoiding safety-critical experts during inference on harmful prompts, and allowing harmful outputs to be elicited.

Given this, the authors propose adding a regularization loss supervising the routing scores to remain similar to the initial aligned model throughout training. This loss is computed on a set of harmful prompts, which need not feature in the training set provided by the attacker.

Algorithmically, the authors propose a bi-level greedy optimization setup that alternates between minimizing the SFT loss and their routing regularization loss.

SafeMoE on a range of mixture-of-experts LLMs and tasks by simulating harmful fine-tuning (mixing standard task data with a small set of harmful prompts) and measuring both safety and utility. Utility is assessed on SAMSum summarization (ROUGE-1) and SQL generation (exact match), safety is judged on JailbreakBench via an automated moderator, and comparisons are made against fine-tuning-stage baselines (SafeInstr, SaLoRA) and post-hoc edits (Antidote, SafeDelta). Across small and large MoE models, SafeMoE consistently cuts harmfulness dramatically while preserving or slightly improving task performance, remains effective under full-parameter fine-tuning with minimal training overhead, and shows robustness in sensitivity studies (varying harmful sample count, ratio, and regularization temperature). Analyses of training dynamics further link its gains to reduced routing drift - especially in upper layers—helping argue that the safety improvements stem from stabilizing expert selection rather than sacrificing capability.

**Strengths:**

- Presentation: the paper is clearly written and well-presented, which facilitated understanding its motivation, main proposed method and experiment results.
- Strong results: the results indicate SafeMOE achieves significantly lower harmfulness scores than other methods, while preserving fine-tuning accuracy.
- Low overhead: SafeMOE can be run at a low compute overhead by only sporadically optimizing the routing regularization loss. This makes it cheap for practitioners to adopt the method, making it more likely to become relevant for the community.

**Weaknesses:**

- Correlation vs. causation
  - The author’s proposed correlation analysis in Section 3 seems to me to be confounded, and to conflate correlation with causation. The authors track routing drift and harmfulness score through fine-tuning, and observe that higher routing drift is associated with a higher harmfulness score. However, given that the harmfulness score of the final model is known to be higher than of the initial model (based on e.g. prior work), there are several metrics one could track that would also increase during training, even though they might not causally affect the harmfulness score. For instance, the distance between initial and final parameters likely increases during fine-tuning as well, and yet one would not conclude from this that “the further the initial parameters are from the final parameters, the more harmful the final model would be”.
  - In this case, since the author’s hypothesized mechanism is routing drift, I believe a more appropriate and controlled experiment would be to freeze the routing weights to be those of the initial model, and evaluate the harmfulness score during fine-tuning when using those frozen routing weights at inference time. If the harmfulness score becomes significantly lower, this would give more direct and less confounded evidence to the authors’ claim.
- Lack of stopgrad in $L_{reg}$ definition
  - My understanding, from reading the code (lines 866–899 of `safemoe_trainer.py`) is that, in the routing regularization function, no stopgrad is applied to the input of the routing layer. Given this, my understanding is that the gradients of the routing regularization loss would propagate to the the residual stream vectors, and hence to all previous layers of the model; not just to the routing.
  - Given this, it is not clear to me that all this regularization loss is doing is enforcing the routing to stay the same. For instance, it is possible that the effect of the method is to cause activations to remain largely the same, which would lead to a different conclusion as to how SafeMOE achieves its results.
  - It would help put the authors’ contributions into perspective if we had an ablation where stopgrad is applied to the activations fed into the routing module. If the results are similar to those of SafeMOE, we can be reassured that the reason SafeMOE works is in fact due to avoiding routing drift. If not, it might be that a different effect is at play, and it would be important to understand what that is.
  - Conversely, another potentially relevant ablation would be to use a regularization loss that simply supervises residual stream activations to stay the same for the harmful prompts (e.g. via an $L^2$ loss). My prior is that this would harm fine-tuning accuracy a lot more than SafeMOE does, but it would be interesting to check, in case the authors can run such an experiment easily (e.g. on a small model).

**Questions:**

- Why would it be prohibitive to optimize $L_{\text{reg}}$ together with $L_{\text{sft}}$, for instance using a Monte Carlo estimate of the $L_{\text{reg}}$ gradient? Mechanically, this would correspond to sampling a batch from $D_h$ at each optimization step. Is this because the sequence batch size is small, so that adding even a handful of harmful prompts to the batch would be prohibitive?
- When defining routing drift and $L_{\text{reg}}$, what is the authors’ intuition for why the reference measure should come from the current model checkpoint ($w$) instead of from the aligned model ($w_\text{align}$)? I.e. why do you use $\mathrm{KL}(\text{aligned} || \text{current})$ instead of $\mathrm{KL}(\text{current} || \text{aligned})$? Do you expect this should make a difference, or should it not matter much? My understanding is that the more common form in KL regularization for fine-tuning is to use the latter.

I will increase my score if the concerns regarding causal claims in section 3 and the absence of stopgrad in the regularization loss definition are addressed. The main things I would like to see in these directions are the ablations highlighted in the Weaknesses section.

---

> ### Author Response · Authors · 2025-11-21
> **Rebuttal by Authors (1/2)**
>
> We sincerely appreciate the reviewer’s constructive feedback, especially the suggestion to include controlled experiments to strengthen our claims. We provide our detailed responses and additional experimental results below. We have uploaded the revised manuscript that incorporates these responses and results, with corresponding updates specified for each question. We hope these clarifications address the concerns raised. If you have any further questions or comments, we would be grateful to receive them.
>
> ### **1. Controlled analysis of frozen routing weights**
>
> > Revision in *Section 3*, *Figure 4*
>
> To assess the significance of routing weights in maintaining safety, we conducted an experiment using the OLMoE model in which the routing weights (i.e., the outputs of the gating networks) were frozen. Specifically, for the fine-tuned model, we replaced the routing weights for each input with those from the initial safety-aligned model at inference time and measured the harmfulness score over multiple training steps. If the harmfulness scores remained lower than those of the original fine-tuned model, it would indicate that preserving the initial routing decisions (i.e., the routing weights) plays a critical role in reducing harmfulness during fine-tuning.
>
> As shown in the results of the OLMoE model on the SAMSum task below, the measured harmfulness scores are indeed substantially lower than those of the fine-tuned baseline. This confirms that routing weights are crucial to MoE safety and that constraining safety routing drift during fine-tuning is a sound strategy for mitigating HFT attacks. We have revised our manuscript by including these experimental results and corresponding discussions.
>
> | Fine-tuning step | HS (frozen routing weights) | HS (original fine-tuning) |
> |------------------|--------------------------------------------|---------------------------------|
> | 50               | 8.0                                        | 11.0                            |
> | 100              | 10.0                                       | 35.0                            |
> | 150              | 19.0                                       | 53.0                            |
> | 200              | 17.0                                       | 59.0                            |
> | 250              | 13.0                                       | 60.0                            |
> | 300              | 15.0                                       | 61.0                            |
> | 350              | 14.0                                       | 63.0                            |
> | 400              | 16.0                                       | 61.0                            |
> | 450              | 16.0                                       | 61.0                            |
> | 500              | 16.0                                       | 60.0                            |
> *HS: Harmfulness score
>
> ### **2. Stopgrad in $\mathcal{L}_{reg}$ definition**
> > Revision in *Section 5.5*, *Table 4*
>
> The routing weights of an MoE LLM are generated by the gating networks, which take the residual vector as input. Therefore, the residual vector also affects the routing weights, and our method is designed to preserve these final routing weights (i.e., routing decisions). Allowing gradient flows (through the gating networks) to the residual vector is thus consistent with the intended design of SafeMoE.
>
> We appreciate the reviewer’s suggestion and conducted an additional ablation study on gradient flows for isolating and understanding the contribution of our routing preservation approach.
> An MoE module has two forward paths (input → layer → output): 1) *residual vector → gating network → routing weights*, which determines routing decisions, and 2) *residual vector → expert layers → expert output vectors*, which produces expert representations.
>
> In our ablation, we block gradient flows from the expert output vectors back into the residual vector through the second path. This enables us to separately examine the effect of routing preservation without interference from residual vector updates induced by the previous transformer layers and the expert layers. The results below show that this ablation setting still achieves strong defense performance, supporting the validity of our claims.
>
> We included these experimental results and discussions in the revised version.
>
> | Method             | Fine-tuning accuracy | Harmfulness score |
> |--------------------|----------------------|-------------------|
> | Fine-tuning (w/o defense)       | 49.3                 | 62.0              |
> | SafeMoE            | 48.9                 | 5.0               |
> | SafeMoE (ablation) | 48.2                 | 18.0              |

---

> > ### Author Response · Authors · 2025-11-21
> > **Rebuttal by Authors (2/2)**
> >
> > ### **3. Rationale behind bi-level greedy optimization**
> > We already explored joint optimization strategies, including a Monte Carlo estimation approach that samples harmful instructions for drift regularization. However, in larger models (> 16B), it is challenging due to memory constraints and substantial computational overhead in our environment with a single A100 GPU. Our analysis below shows that the Monte Carlo-based approach incurs significant execution-time overhead. For this reason, we propose a highly scalable yet sufficiently effective alternative for SafeMoE.
> >
> > | Defense               | Fine-tuning accuracy | Harmfulness score | Extra time (%)   |
> > |-----------------------|----------------------|-------------------|------------------|
> > | SafeMoE               | 48.9                 | 5.0               | 17.26s (2.13%)   |
> > | SafeMoE (Monte Carlo) | 48.1                 | 0                 | 529.78s (65.49%) |
> >
> > ### **4. Calculation of KL divergence**
> >
> > > Revision in *Section 3*
> >
> > We used the routing weights from the aligned model as the reference distribution and computed $D_{KL}(reference \text{(i.e., aligned)} || input \text{(i.e., current to fine-tune)} )$ in accordance with the standard definition of KL divergence [R1]. We have confirmed that our implementation is consistent with this formulation. We will explicitly clarify this point in the revised version to avoid any confusion. If you find any issues with our explanation, we would be grateful if you could let us know.
> >
> > [R1] Kullback, Solomon, and Richard A. Leibler. “On Information and Sufficiency.” The annals of mathematical statistics 22.1 (1951): 79-86.

---

> ### Author Response · Authors · 2025-11-28
>
> Dear Reviewer
>
> We sincerely appreciate your constructive feedback once again. We have made substantial improvements based on your valuable suggestions. If possible, we would be grateful for any further comments. Please let us know if you feel that additional clarification or revisions are needed.
>
> Sincerely,\
> The authors

---

### Author Response · Authors · 2025-11-29
**Comment to AC After Reverting**

Dear Area Chair,

We have addressed all reviewers' questions and concerns in our rebuttal and posted the requested experimental results. Also, the revised paper contains all the required changes.

For your information, Reviewer Maxq already updated their merit score to 8 during the rebuttal period. Reviewer Ysqm and Reviewer 6cTn explicitly stated that they would raise their scores once their concerns were addressed. Although we believe we have also thoroughly handled their concerns, as well as those of the other reviewer (Reviewer YmtJ), we have not received any further engagement from these three reviewers.

If you have any concerns or questions regarding our response, please let us know.

We truly appreciate your time and effort throughout the process.

Sincerely,\
The authors

---

### Meta-Review · Area_Chair_bpbq · 2026-01-04

**Summary:**

This paper is the first to systematically study the problem of harmful fine-tuning on Mixture-of-Experts (MoE) models. The authors discover that harmful fine-tuning causes the routing weights of MoE models to drift away from the aligned model, a phenomenon they term "safety routing drift." They propose a simple and effective regularization method, SafeMoE, to mitigate this drift and enhance the safety of fine-tuned MoE LLMs. The paper provides comprehensive experiments on various MoE models and tasks, demonstrating that SafeMoE can significantly reduce harmfulness while maintaining task performance.

The work is novel, the methodology is sound, and the empirical results are strong. The paper is also exceptionally well-written and easy to follow. The authors have done an outstanding job of responding to the reviewers' feedback and have further strengthened the paper through their rebuttal. Therefore, I recommend acceptance for this paper.

**Reviewer Concerns:**

* *Missing Baselines:* One reviewer (6cTn) suggested comparing with another recent method (Lisa). This has been addressed.

* *Threat Model:* One reviewer (Maxq) questioned the robustness of SafeMoE against extreme adversarial attacks. This has been addressed.

* *Mechanistic Explanation:* One reviewer (Maxq) requested a more detailed explanation for the rapid increase in harmfulness during the early stages of fine-tuning. This has been addressed.

**Reviewer Scores:**

* Reviewer Ysqm: The concern of this reviewer has been partially addressed, and he/she may increase the score to 6.

* Reviewer Maxq: This reviewer has already indicated he/she will raise their score.

* Reviewer YmtJ: The concern of this reviewer has been addressed, and he/she will likely increase the score to 4 or 6.

* Reviewer 6cTn: While this reviewer maintained their score of 6, their comments were very positive, and it is possible they would have also raised their score after seeing the final version of the paper.

---

### Decision · Program_Chairs · 2026-01-26

Accept (Poster)